# ADAPTIVE SINGLE-PASS STOCHASTIC GRADIENT DESCENT IN INPUT SPARSITY TIME

## ABSTRACT

We study sampling algorithms for variance reduction methods for stochastic optimization. Although stochastic gradient descent (SGD) is widely used for large scale machine learning, it sometimes experiences slow convergence rates due to the high variance from uniform sampling. In this paper, we introduce an algorithm that approximately samples a gradient from the optimal distribution for a common finite-sum form with $n$ terms, while just making a single pass over the data, using input sparsity time, and $\tilde{\mathcal{O}}(Td)$ space. Our algorithm can be implemented in big data models such as the streaming and distributed models. Moreover, we show that our algorithm can be generalized to approximately sample Hessians and thus provides variance reduction for second-order methods as well. We demonstrate the efficiency of our algorithm on large-scale datasets.

## 1 INTRODUCTION

There has recently been tremendous progress in variance reduction methods for stochastic gradient descent (SGD) methods for the standard convex *finite-sum form* optimization problem $\min\limits_{\mathbf{x} \in \mathbb{R}^d} F(\mathbf{x}) := \frac{1}{n} \sum_{i=1}^{n} f_i(\mathbf{x})$, where $f_1, \ldots, f_n : \mathbb{R}^d \to \mathbb{R}$ is a set of convex functions that commonly represent loss functions. Whereas gradient descent (GD) performs the update rule $\mathbf{x}_{t+1} = \mathbf{x}_t - \eta_t \nabla F(\mathbf{x}_t)$ on the iterative solution $\mathbf{x}_t$ at iterations $t = 1, 2, \ldots$, SGD (Robbins & Monro, 1951; Nemirovsky & Yudin, 1983; Nemirovski et al., 2009) picks $i_t \in [n]$ in iteration $t$ with probability $p_{i_t}$ and performs the update rule $\mathbf{x}_{t+1} = \mathbf{x}_t - \frac{\eta_t}{n p_{i_t}} \nabla f_{i_t}(\mathbf{x}_t)$, where $\nabla f_{i_t}$ is the gradient (or a subgradient) of $f_{i_t}$ and $\eta_t$ is some predetermined learning rate. Effectively, training example $i_t$ is sampled with probability $p_{i_t}$ and the model parameters are updated using the selected example.

The SGD update rule only requires the computation of a single gradient at each iteration and provides an unbiased estimator to the full gradient, compared to GD, which evaluates $n$ gradients at each iteration and is prohibitively expensive for large $n$. However, since SGD is often performed with uniform sampling so that the probability $p_{i,t}$ of choosing index $i \in [n]$ at iteration $t$ is $p_{i,t} = \frac{1}{n}$ at all times, the variance introduced by the randomness of sampling a specific vector function can be a bottleneck for the convergence rate of the iterative process. Thus the subject of variance reduction beyond uniform sampling has been well-studied in recent years (Roux et al., 2012; Johnson & Zhang, 2013; Defazio et al., 2014; Reddi et al., 2015; Zhao & Zhang, 2015; Daneshmand et al., 2016; Needell et al., 2016; Stich et al., 2017; Johnson & Guestrin, 2018; Katharopoulos & Fleuret, 2018; Salehi et al., 2018; Qian et al., 2019).

A common technique to reduce variance is importance sampling, where the probabilities $p_{i,t}$ are chosen so that vector functions with larger gradients are more likely to be sampled. Thus for $\text{Var}(\mathbf{v}) := \mathbb{E}\left[\|\mathbf{v}\|_2^2\right] - \|\mathbb{E}[\mathbf{v}]\|_2^2$, for a random vector $\mathbf{v}$, then $p_{i,t} = \frac{1}{n}$ for uniform sampling implies

$$\sigma_t^2 = \text{Var}\left(\frac{1}{n p_{i_t,t}} \nabla f_{i_t}\right) = \frac{1}{n^2}\left(n \sum_{i=1}^{n} \|\nabla f_i(\mathbf{x}_t)\|^2 - n^2 \cdot \|\nabla F(\mathbf{x}_t)\|^2\right),$$

whereas importance sampling with $p_{i,t} = \frac{\|\nabla f_i(\mathbf{x}_t)\|}{\sum_{j=1}^n \|\nabla f_j(\mathbf{x}_t)\|}$ gives

$$\sigma_t^2 = \mathrm{Var}\left(\frac{1}{np_{i_t,t}}\nabla f_{i_t}\right) = \frac{1}{n^2}\left(\left(\sum_{i=1}^n \|\nabla f_i(\mathbf{x}_t)\|\right)^2 - n^2 \cdot \|\nabla F(\mathbf{x}_t)\|^2\right),$$

which is at most $\frac{1}{n^2}\left(n\sum\|\nabla f_i(\mathbf{x}_t)\|^2 - n^2 \cdot \|\nabla F(\mathbf{x}_t)\|^2\right)$, by the Root-Mean Square-Arithmetic Mean Inequality, and can be significantly less. Hence the variance at each step is reduced, possibly substantially, e.g., Example 1.3 and Example 1.4, by performing importance sampling instead of uniform sampling. In fact, it follows from the Cauchy-Schwarz inequality that the above importance sampling probability distribution is the optimal distribution for variance reduction. However, computing the probability distribution for importance sampling requires computing the gradients in each round, which is too expensive in the first place.

**Second-Order Methods.** Although first-order methods such as SGD are widely used, they do sometimes have issues such as sensitivity to the choice of hyperparameters, stagnation at high training errors, and difficulty in escaping saddle points. By considering second-order information such as curvature, second-order optimization methods are known to be robust to several of these issues, such as ill-conditioning. For example, Newton's method can achieve a locally super-linear convergence rate under certain conditions, independent of the problem. Although naïve second-order methods are generally too slow compared to common first-order methods, stochastic Newton-type methods such as Gauss-Newton have shown to be scalable in the scientific computing community (Roosta-Khorasani et al., 2014; Roosta-Khorasani & Mahoney, 2016a;b; Xu et al., 2019; 2020).

**Our Contributions.** We give a time efficient algorithm that *provably* approximates the optimal importance sampling using a small space data structure. Remarkably, our data structure can be implemented in big data models such as the streaming model, which just takes a single pass over the data, and the distributed model, which requires just a single round of communication between parties holding each loss function. For $\nabla F = \frac{1}{n}\sum \nabla f_i(\mathbf{x})$, where each $\nabla f_i = f(\langle \mathbf{a}_i, \mathbf{x}\rangle) \cdot \mathbf{a}_i$ for some polynomial $f$ and vector $\mathbf{a}_i \in \mathbb{R}^d$, let $\mathrm{nnz}(\mathbf{A})$ be the number of nonzero entries of $\mathbf{A} := \mathbf{a}_1 \circ \ldots \circ \mathbf{a}_n$[1]. Thus for $T$ iterations, where $d \ll T \ll n$, GD has runtime $\tilde{\mathcal{O}}(T \cdot \mathrm{nnz}(\mathbf{A}))$ while our algorithm has runtime $T \cdot \mathrm{poly}(d, \log n) + \tilde{\mathcal{O}}(\mathrm{nnz}(\mathbf{A}))$, where we use $\tilde{\mathcal{O}}(\cdot)$ to suppress polylogarithmic terms.

**Theorem 1.1** *Let $\nabla F = \frac{1}{n}\sum \nabla f_i(\mathbf{x})$, where each $\nabla f_i = f(\langle \mathbf{a}_i, \mathbf{x}\rangle) \cdot \mathbf{a}_i$ for some polynomial $f$ and vector $\mathbf{a}_i \in \mathbb{R}^d$ and let $\mathrm{nnz}(\mathbf{A})$ be the number of nonzero entries of $\mathbf{A} := \mathbf{a}_1 \circ \ldots \circ \mathbf{a}_n$. For $d \ll T \ll n$, there exists an algorithm that performs $T$ steps of SGD and at each step samples a gradient within a constant factor of the optimal probability distribution. The algorithm requires a single pass over $\mathbf{A}$ and uses $\tilde{\mathcal{O}}(\mathrm{nnz}(\mathbf{A}))$ pre-processing time and $\tilde{\mathcal{O}}(Td)$ space.*

Theorem 1.1 can be used to immediately obtain improved convergence guarantees for a class of functions whose convergence rate depends on the variance $\sigma_t^2$, such as $\mu$-smooth functions or strongly convex functions. Recall that SGD offers the following convergence guarantees for smooth functions:

**Theorem 1.2** *(Nemirovski et al., 2009; Meka, 2017) Let $F$ be a $\mu$-smooth convex function and $\mathbf{x}_{\mathsf{opt}} = \arg\min F(\mathbf{x})$. Let $\sigma^2$ be an upper bound for the variance of the unbiased estimator across all iterations and $\overline{\mathbf{x}_k} = \frac{\mathbf{x}_1 + \ldots + \mathbf{x}_k}{k}$. Let each step-size $\eta_t$ be $\eta \leq \frac{1}{\mu}$. Then for SGD with initial position $\mathbf{x}_0$,*

$$\mathbb{E}\left[F(\overline{\mathbf{x}_k}) - F(\mathbf{x}_{\mathsf{opt}})\right] \leq \frac{1}{2\eta k}\|\mathbf{x}_0 - \mathbf{x}_{\mathsf{opt}}\|_2^2 + \frac{\eta\sigma^2}{2},$$

*so that $k = \mathcal{O}\left(\frac{1}{\epsilon^2}\left(\sigma^2 + \mu\|\mathbf{x}_0 - \mathbf{x}_{\mathsf{opt}}\|_2^2\right)^2\right)$ iterations suffices to obtain an $\epsilon$-approximate optimal value by setting $\eta = \frac{1}{\sqrt{k}}$.*

In the convergence guarantees of Theorem 1.2, we obtain a constant factor approximation to the variance $\sigma = \sigma_{opt}$ from optimal importance sampling, which can be significantly better than the

---

[1]We use the notation $\mathbf{a} \circ \mathbf{b}$ to denote the vertical concatenation $\begin{bmatrix}\mathbf{a}\\\mathbf{b}\end{bmatrix}$.

variance $\sigma = \sigma_{uniform}$ from uniform sampling in standard SGD. We first show straightforward examples where uniform sampling an index performs significantly worse than importance sampling. For example, if $\nabla f_i(\mathbf{x}) = \langle \mathbf{a}_i, \mathbf{x} \rangle \cdot \mathbf{a}_i$, then for $\mathbf{A} = \mathbf{a}_1 \circ \ldots \circ \mathbf{a}_n$:

**Example 1.3** *When the nonzero entries of the input $\mathbf{A}$ are concentrated in a small number of vectors $\mathbf{a}_i$, uniform sampling will frequently sample gradients that are small and make little progress, whereas importance sampling will rarely do so. In an extreme case, the $\mathbf{A}$ can contain exactly one nonzero vector $\mathbf{a}_i$ and importance sampling will always output the full gradient whereas uniform sampling will only find the nonzero row with probability $\frac{1}{n}$.*

**Example 1.4** *It may be that all rows of $\mathbf{A}$ have large magnitude, but $\mathbf{x}$ is nearly orthogonal to most of the rows of $\mathbf{A}$ and heavily in the direction of row $\mathbf{a}_r$. Then $\langle \mathbf{a}_i, \mathbf{x} \rangle \cdot \mathbf{a}_i$ is small in magnitude for most $i$, but $\langle \mathbf{a}_r, \mathbf{x} \rangle \cdot \mathbf{a}_r$ is large so uniform sampling will often output small gradients while importance sampling will output $\langle \mathbf{a}_r, \mathbf{x} \rangle \cdot \mathbf{a}_r$ with high probability.*

Thus Example 1.3 shows that naïve SGD with uniform sampling can suffer up to a multiplicative $n$ factor loss in the convergence rate of Theorem 1.2 compared to that of SGD with importance sampling whereas Example 1.4 shows a possible additive $n$ factor loss.

Unlike a number of previous variance reduction methods, we do not require distributional assumptions (Bouchard et al., 2015; Frostig et al., 2015; Gopal, 2016; Jothimurugesan et al., 2018) or offline access to the data (Roux et al., 2012; Johnson & Zhang, 2013; Defazio et al., 2014; Reddi et al., 2015; Zhao & Zhang, 2015; Daneshmand et al., 2016; Needell et al., 2016; Stich et al., 2017; Johnson & Guestrin, 2018; Katharopoulos & Fleuret, 2018; Salehi et al., 2018; Qian et al., 2019). On the other hand, for applications such as neural nets in which the parameters in the loss function can change, we can use a second-order approximation for a number of iterations, then reread the data to build a new second-order approximation when necessary.

We complement our main theoretical result with empirical evaluations comparing our algorithm to SGD with uniform sampling for logistic regression on the a9a Adult dataset collected by UCI and retrieved from LibSVM (Chang & Lin, 2011). Our evaluations demonstrate that for various step-sizes, our algorithm has significantly better performance than uniform sampling across both the number of SGD iterations and surprisingly, wall-clock time.

We then show that our same framework can also be reworked to approximate importance sampling for the Hessian, thereby performing variance reduction for second-order optimization methods. (Xu et al., 2016) reduce the bottleneck of many second-order optimization methods to the task of sampling $s$ rows of $\mathbf{A} = \mathbf{a}_1 \circ \ldots \circ \mathbf{a}_n$ so that a row $\mathbf{a}_i$ is sampled with probability $\frac{\left\| f(\langle \mathbf{a}_i, \mathbf{x} \rangle) \cdot \mathbf{a}_i^\top \mathbf{a}_i \right\|_F^2}{\sum_{i=1}^n \left\| f(\langle \mathbf{a}_i, \mathbf{x} \rangle) \cdot \mathbf{a}_i^\top \mathbf{a}_i \right\|_F^2}$, for some fixed function $f$ so that the Hessian $\mathbf{H}$ has the form $\mathbf{H} := \nabla^2 F = \frac{1}{n} \sum \nabla f(\langle \mathbf{a}_i, \mathbf{x} \rangle) \mathbf{a}_i^\top \mathbf{a}_i$. (Xu et al., 2016) show that this finite-sum form arises frequently in machine learning problems such as logistic regression with least squares loss.

**Theorem 1.5** *Let $\nabla^2 F = \frac{1}{n} \sum \nabla f_i(\mathbf{x})$, where each $\nabla f_i = f(\langle \mathbf{a}_i, \mathbf{x} \rangle) \cdot \mathbf{a}_i^\top \mathbf{a}_i$ for some polynomial $f$ and vector $\mathbf{a}_i \in \mathbb{R}^d$ and let $\mathrm{nnz}(\mathbf{A})$ be the number of nonzero entries of $\mathbf{A} := \mathbf{a}_1 \circ \ldots \circ \mathbf{a}_n$. For $d \ll T \ll n$, there exists an algorithm that subsamples $T$ Hessians within a constant factor of the optimal probability distribution. The algorithm requires a single pass over $\mathbf{A}$ and uses $\tilde{\mathcal{O}}\left(\mathrm{nnz}(\mathbf{A})\right)$ pre-processing time and $\tilde{\mathcal{O}}\left(Td\right)$ space.*

## 2 SGD ALGORITHM

We first introduce a number of algorithms that will be used in our final SGD algorithm, along with their guarantees. We defer all formal proofs to the appendix.

$L_2$ **polynomial inner product sketch.** For a fixed polynomial $f$, we first require a constant-factor approximation to $\left\| \sum_{i=1}^n f(\langle \mathbf{a}_i, \mathbf{x} \rangle) \cdot \mathbf{a}_i \right\|_2$ for any query $\mathbf{x} \in \mathbb{R}^d$; we call such an algorithm an $L_2$ *polynomial inner product sketch* and give such an algorithm with the following guarantee:

**Theorem 2.1** *For a fixed $\epsilon > 0$ and polynomial $f$, there exists a data structure ESTIMATOR that outputs a $(1+\epsilon)$-approximation to $\sum_{i=1}^n \left\| f(\langle \mathbf{a}_i, \mathbf{x} \rangle) \cdot \mathbf{a}_i \right\|_2^2$ for any query $\mathbf{x} \in \mathbb{R}^d$. The data structure*

*requires a single pass over* $\mathbf{A} = \mathbf{a}_1 \circ \ldots \circ \mathbf{a}_n$ *(possibly through turnstile updates[2]), can be built in* $\tilde{O}\left(\mathrm{nnz}(\mathbf{A}) + \frac{d}{\epsilon^2}\right)$ *time and* $\tilde{O}\left(\frac{d}{\epsilon^2}\right)$ *space, uses query time* $\mathrm{poly}\left(d, \frac{1}{\epsilon}, \log n\right)$*, and succeeds with probability* $1 - \frac{1}{\mathrm{poly}(n)}$*.*

The $L_2$ polynomial inner product sketch ESTIMATOR is a generalization of AMS variants Alon et al. (1999); Mahabadi et al. (2020) and is simple to implement. For intuition, observe that for $d = 1$ and the identity function $f$, the matrix $\mathbf{A} \in \mathbb{R}^{n \times d}$ reduces to a vector of length $n$ so that estimating $\sum_{i=1}^{n} \|f(\langle \mathbf{a}_i, \mathbf{x} \rangle) \cdot \mathbf{a}_i\|_2^2$ is just estimating the squared norm of a vector in sublinear space. For a degree $p$ polynomial $f$, ESTIMATOR generates random sign matrices $\mathbf{S}_0, \mathbf{S}_1, \ldots, \mathbf{S}_p$ with $\tilde{O}\left(\frac{1}{\epsilon^2}\right)$ rows and maintains $\mathbf{S}_0\mathbf{A}, \ldots, \mathbf{S}_p\mathbf{A}$. To estimate $\sum_{i=1}^{n} \|\alpha_q \cdot (\langle \mathbf{a}_i, \mathbf{x} \rangle)^q \cdot \mathbf{a}_q\|_2^2$ for an integer $q \in [0, p]$ and scalar $\alpha_q$ on a given query $\mathbf{x}$, ESTIMATOR creates the $q$-fold tensor $\mathbf{Y} = \mathbf{y}^{\otimes q}$ for each row $\mathbf{y}$ of $\mathbf{S}_q\mathbf{A}$ and the $(q-1)$-fold tensor $\mathbf{X} = \mathbf{x}^{\otimes(q-1)}$. Note that $\mathbf{X}$ and $\mathbf{Y}$ can be refolded into dimensions $\mathbb{R}^{d^{q-1}}$ and $\mathbb{R}^{d \times d^{q-1}}$ so that $\mathbf{YX} \in \mathbb{R}^d$ and $\|\alpha_q \cdot \mathbf{YX}\|_2^2$ is an unbiased estimator of $\sum_{i=1}^{n} \|\alpha_q \cdot (\langle \mathbf{a}_i, \mathbf{x} \rangle)^q \cdot \mathbf{a}_q\|_2^2$. We give this algorithm in full in Algorithm 1. Thus, taking the average over $\mathcal{O}\left(\frac{1}{\epsilon^2}\right)$ instances of the sums of the tensor products for rows $\mathbf{y}$ across the sketches $\mathbf{S}_0\mathbf{A}, \ldots, \mathbf{S}_p\mathbf{A}$ gives a $(1 + \epsilon)$-approximation to $\sum_{i=1}^{n} \|f(\langle \mathbf{a}_i, \mathbf{x} \rangle) \cdot \mathbf{a}_i\|_2^2$ with constant probability. The success probability of success can then be boosted to $1 - \frac{1}{\mathrm{poly}(n)}$ by taking the median of $\mathcal{O}(\log n)$ such outputs.

---

**Algorithm 1** Basic algorithm ESTIMATOR that outputs $(1 + \epsilon)$-approximation to $\sum_{i=1}^{n} \|(\langle \mathbf{a}_i, \mathbf{x} \rangle)^p \cdot \mathbf{a}_i\|_2^2$, where $\mathbf{x}$ is a post-processing vector

---

**Input:** Matrix $\mathbf{A} = \mathbf{a}_1 \circ \ldots \circ \in \mathbb{R}^{n \times d}$, post-processing vector $\mathbf{x} \in \mathbb{R}^d$, integer $p \geq 0$, constant parameter $\epsilon > 0$.
**Output:** $(1 + \epsilon)$-approximation to $\sum_{i=1}^{n} \|(\langle \mathbf{a}_i, \mathbf{x} \rangle)^p \cdot \mathbf{a}_i\|_2^2$.
1: $r \leftarrow \Theta(\log n)$ with a sufficiently large constant.
2: $b \leftarrow \Omega\left(\frac{1}{\epsilon^2}\right)$ with a sufficiently large constant.
3: Let $\mathcal{T}$ be an $r \times b$ table of buckets, where each bucket stores an $\mathbb{R}^d$ vector, initialized to the zeros vector.
4: Let $s_i \in \{-1, +1\}$ be 4-wise independent for $i \in [n]$.
5: Let $h_i : [n] \to [b]$ be 4-wise independent for $i \in [r]$.
6: Let $\mathbf{u}_{i,j}$ be the all zeros vector for each $i \in [r], j \in [b]$.
7: **for** each $j = 1$ to $n$ **do**
8:     **for** each $i = 1$ to $r$ **do**
9:         Add $s_j\mathbf{a}_j$ to the vector in bucket $h_i(j)$ of row $i$.
10: Let $\mathbf{v}_{i,j}$ be the vector in row $i$, bucket $j$ of $\mathcal{T}$ for $i \in [r], j \in [b]$.
11: **Process x:**
12: **for** $i \in [r], j \in [b]$ **do**
13:     $\mathbf{u}_{i,j} \leftarrow \mathbf{v}_{i,j}^{\otimes p}\mathbf{x}^{\otimes(p-1)}$
14: **return** $\mathrm{median}_{i \in [r]} \frac{1}{b} \sum_{j \in [b]} \|\mathbf{u}_{i,j}\|_2^2$.

---

$L_2$ **polynomial inner product sampler.** Given a matrix $\mathbf{A} = \mathbf{a}_1 \circ \ldots \circ \mathbf{a}_n \in \mathbb{R}^{n \times d}$ and a fixed function $f$, a data structure that takes query $\mathbf{x} \in \mathbb{R}^d$ and outputs an index $i \in [n]$ with probability roughly

$$\frac{\|f(\langle \mathbf{a}_i, \mathbf{x} \rangle) \cdot \mathbf{a}_i\|_2^2}{\sum_{i=1}^{n} \|f(\langle \mathbf{a}_i, \mathbf{x} \rangle) \cdot \mathbf{a}_i\|_2^2}$$

is called an $L_2$ *polynomial inner product sampler*. We give such a data structure in Section A.1:

**Theorem 2.2** *For a fixed $\epsilon > 0$ and polynomial $f$, there exists a data structure* SAMPLER *that takes any query* $\mathbf{x} \in \mathbb{R}^d$ *and outputs an index* $i \in [n]$ *with probability* $\frac{(1 \pm \epsilon) \cdot \|f(\langle \mathbf{a}_i, \mathbf{x} \rangle) \cdot \mathbf{a}_i\|_2^2}{\sum_{i=1}^{n} \|f(\langle \mathbf{a}_i, \mathbf{x} \rangle) \cdot \mathbf{a}_i\|_2^2} + \frac{1}{\mathrm{poly}(n)}$*, along with a vector* $\mathbf{u} := f(\langle \mathbf{a}_i, \mathbf{x} \rangle) \cdot \mathbf{a}_i + \mathbf{v}$*, where* $\mathbb{E}[\mathbf{v}] = 0$ *and* $\|\mathbf{v}\|_2 \leq \epsilon \cdot \|f(\langle \mathbf{a}_i, \mathbf{x} \rangle) \cdot \mathbf{a}_i\|_2$*. The data*

---

[2]Turnstile updates are defined as sequential updates to the entries of $\mathbf{A}$.

*structure requires a single pass over* $\mathbf{A} = \mathbf{a}_1 \circ \ldots \circ \mathbf{a}_n$ *(possibly through turnstile updates), can be built in* $\tilde{\mathcal{O}}\left(\mathrm{nnz}(\mathbf{A}) + \frac{d}{\epsilon^2}\right)$ *time and* $\tilde{\mathcal{O}}\left(\frac{d}{\epsilon^2}\right)$ *space, uses query time* $\mathrm{poly}\left(d, \frac{1}{\epsilon}, \log n\right)$, *and succeeds with probability* $1 - \frac{1}{\mathrm{poly}(n)}$.

We remark that $T$ independent instances of SAMPLER provide an oracle for $T$ steps of SGD with importance sampling, but the overall runtime would be $T \cdot \mathrm{nnz}(\mathbf{A})$ so it would be just as efficient to run $T$ iterations of GD. The subroutine SAMPLER is significantly more challenging to describe and analyze, so we defer its discussion to Section A.1, though it can be seen as a combination of ESTIMATOR and a generalized CountSketch Charikar et al. (2004); Nelson & Nguyen (2013); Mahabadi et al. (2020) variant and is nevertheless relatively straightforward to implement.

**Leverage score sampler.** Although SAMPLER outputs a (noisy) vector according to the desired probability distribution, we also require an algorithm that automatically does this for indices $i \in [n]$ that are likely to be sampled multiple times across the $T$ iterations. Equivalently, we require explicitly storing the rows with high leverage scores, but we defer the formal discussion and algorithmic presentation to Section A.2. For our purposes, the following suffices:

**Theorem 2.3** *There exists an algorithm* LEVERAGE *that returns all indices* $i \in [n]$ *such that* $\frac{(1 \pm \epsilon) \cdot \|f(\langle \mathbf{a}_i, \mathbf{x} \rangle) \cdot \mathbf{a}_i\|_2^2}{\sum_{i=1}^n \|f(\langle \mathbf{a}_i, \mathbf{x} \rangle) \cdot \mathbf{a}_i\|_2^2} \geq \frac{1}{200 T d}$ *for some* $\mathbf{x} \in \mathbb{R}^n$, *along with a vector* $\mathbf{u}_i := f(\langle \mathbf{a}_i, \mathbf{x} \rangle) \cdot \mathbf{a}_i + \mathbf{v}_i$, *where* $\|\mathbf{v}_i\|_2 \leq \epsilon \cdot \|f(\langle \mathbf{a}_i, \mathbf{x} \rangle) \cdot \mathbf{a}_i\|_2$. *The algorithm requires a single pass over* $\mathbf{A} = \mathbf{a}_1 \circ \ldots \circ \mathbf{a}_n$ *(possibly through turnstile updates), uses* $\tilde{\mathcal{O}}\left(\mathrm{nnz}(\mathbf{A}) + \frac{d^\omega}{\epsilon^2}\right)$ *runtime (where* $\omega$ *denotes the exponent of square matrix multiplication) and* $\tilde{\mathcal{O}}\left(\frac{d}{\epsilon^2}\right)$ *space, and succeeds with probability* $1 - \frac{1}{\mathrm{poly}(n)}$.

## 2.1 SGD ALGORITHM AND ANALYSIS

For the finite-sum optimization problem $\min_{\mathbf{x} \in \mathbb{R}^d} F(\mathbf{x}) := \frac{1}{n} \sum_{i=1}^n f_i(\mathbf{x})$, where each $\nabla f_i = f(\langle \mathbf{a}_i, \mathbf{x} \rangle) \cdot \mathbf{a}_i$, recall that we could simply an instance of SAMPLER as an oracle for SGD with importance sampling. However, naïvely running $T$ SGD steps requires $T$ independent instances, which uses $T \cdot \mathrm{nnz}(\mathbf{A})$ runtime by Theorem 2.2. Thus we use a two level data structure by first implicitly partition the rows of matrix $\mathbf{A} = \mathbf{a}_1 \circ \ldots \circ \mathbf{a}_n$ into $\beta := \Theta(Td)$ buckets $B_1, \ldots, B_\beta$ and creating an instance of ESTIMATOR and SAMPLER for each bucket. The idea is that for a given query $\mathbf{x}_t$ in SGD iteration $t \in [T]$, we first query $\mathbf{x}_t$ to each of the ESTIMATOR data structures to estimate $\sum_{i \in B_j} \|f(\langle \mathbf{a}_i, \mathbf{x} \rangle) \cdot \mathbf{a}_i\|_2^2$ for each $j \in [\beta]$. We then sample index $j \in [\beta]$ among the buckets $B_1, \ldots, B_\beta$ with probability roughly $\frac{\sum_{i \in B_j} \|f(\langle \mathbf{a}_i, \mathbf{x}_t \rangle) \cdot \mathbf{a}_i\|_2^2}{\sum_{i=1}^n \|f(\langle \mathbf{a}_i, \mathbf{x}_t \rangle) \cdot \mathbf{a}_i\|_2^2}$. Once we have sampled index $j$, it would seem that querying the instance SAMPLER corresponding to $B_j$ simulates SGD, since SAMPLER now performs importance sampling on the rows in $B_j$, which gives the correct overall probability distribution for each row $i \in [n]$. Moreover, SAMPLER has runtime proportional to the sparsity of $B_j$, so the total runtime across the $\beta$ instances of SAMPLER is $\tilde{\mathcal{O}}\left(\mathrm{nnz}(\mathbf{A})\right)$.

However, an issue arises when the same bucket $B_j$ is sampled multiple times, as we only create a single instance of SAMPLER for each bucket. We avoid this issue by explicitly accounting for the buckets that are likely to be sampled multiple times. Namely, we show that if $\frac{\|f(\langle \mathbf{a}_i, \mathbf{x}_t \rangle) \cdot \mathbf{a}_i\|_2^2}{\sum_{i=1}^n \|f(\langle \mathbf{a}_i, \mathbf{x}_t \rangle) \cdot \mathbf{a}_i\|_2^2} < \frac{1}{200 T d}$ for all $t \in [T]$ and $i \in [n]$, then by Bernstein's inequality, the probability that no bucket $B_j$ is sampled multiple times is at least $\frac{99}{100}$. Thus we use LEVERAGE to separate all such rows $\mathbf{a}_i$ that violate this property from their respective buckets and explicitly track the SGD steps in which these rows are sampled. We give the algorithm in full in Algorithm 2.

The key property achieved by Algorithm 2 in partitioning the rows and removing the rows that are likely to be sampled multiple times is that each of the SAMPLER instances are queried at most once.

**Lemma 2.4** *With probability at least* $\frac{98}{100}$, *each* $t \in [T]$ *uses a different instance of* SAMPLER$_j$.

**Proof of Theorem 1.1:** Consider Algorithm 2. By Lemma 2.4, each time $t \in [T]$ uses a fresh instance of SAMPLER$_j$, so that independent randomness is used. A possible concern is that each instance ESTIMATOR$_j$ is not using fresh randomness, but we observe that ESTIMATOR procedures

---

**Algorithm 2** Approximate SGD with Importance Sampling

---

**Input:** Matrix $\mathbf{A} = \mathbf{a}_1 \circ \ldots \circ \mathbf{a}_n \in \mathbb{R}^{n \times d}$, parameter $T$ for number of SGD steps.
**Output:** $T$ gradient directions.
 1: **Preprocessing Stage:**
 2: $\beta \leftarrow \Theta(Td)$ with a sufficiently large constant.
 3: Let $h : [n] \to [\beta]$ be a uniformly random hash function.
 4: Let $\mathbf{B}_j$ be the matrix formed by the rows $\mathbf{a}_i$ of $\mathbf{A}$ with $h(i) = j$, for each $j \in [\beta]$.
 5: Create an instance ESTIMATOR$_j$ and SAMPLER$_j$ for each $\mathbf{B}_j$ with $j \in [\beta]$ with $\epsilon = \frac{1}{2}$.
 6: Run LEVERAGE to find a set $L_0$ of row indices and corresponding (noisy) vectors.
 7: **Gradient Descent Stage:**
 8: Randomly pick starting location $\mathbf{x}_0$
 9: **for** $t = 1$ to $T$ **do**
10:      Let $q_i$ be the output of ESTIMATOR$_j$ on query $\mathbf{x}_{t-1}$ for each $i \in [\beta]$.
11:      Sample $j \in [\beta]$ with probability $p_j = \frac{q_j}{\sum_{i \in [\beta]} q_i}$.
12:      **if** there exists $i \in L_0$ with $h(i) = j$ **then**
13:          Use ESTIMATOR$_j$, LEVERAGE, and SAMPLER$_j$ to sample gradient $\mathbf{w}_t = \widehat{\nabla f_{i_t}(\mathbf{x}_t)}$
14:      **else**
15:          Use SAMPLER$_j$ to sample gradient $\mathbf{w}_t = \widehat{\nabla f_{i_t}(\mathbf{x}_t)}$
16:      $\widehat{p_{i,t}} \leftarrow \frac{\|\mathbf{w}_t\|_2^2}{\sum_{j \in [\beta]} q_j}$
17:      $\mathbf{x}_{t+1} \leftarrow \mathbf{x}_t - \frac{\eta_t}{n\widehat{p_{i,t}}} \cdot \mathbf{w}_t$

---

is only used in sampling a bucket $j \in [\beta]$ as an $L_2$ polynomial inner product sketch; otherwise the sampling uses fresh randomness whereas the sampling is built into each instance of SAMPLER$_j$. By Theorem 2.2, each index $i$ is sampled with probability within a factor 2 of the importance sampling probability distribution. By Theorem 2.1, we have that $\widehat{p_{i,t}}$ is within a factor 4 of the probability $p_{i,t}$ induced by optimal importance sampling SGD. Note that $\mathbf{w}_t = \widehat{\nabla f_i(\mathbf{x}_t)}$ is an unbiased estimator of $\nabla f_i(\mathbf{x}_t)$ and $\|\mathbf{w}_t\|$ is a 2-approximation to $\|\nabla f_i(\mathbf{x}_t)\|$ by Theorem 2.2. Hence, the variance at each time $t \in [T]$ of Algorithm 2 is within a constant factor of the variance $\sigma^2 = \left(\sum \|\nabla f_i(\mathbf{x}_t)\|\right)^2 - \|\nabla F(\mathbf{x}_t)\|^2$ of optimal importance sampling SGD.

By Theorem 2.1, Theorem 2.2, and Theorem 2.3, the preprocessing time is $\tilde{\mathcal{O}}(\text{nnz}(\mathbf{A})) + T \cdot \text{poly}(d, \log n)$ due to the choices of $\epsilon = \mathcal{O}(1)$ and $\beta = \Theta(Td)$, but partitioning the nonzero entries of $\mathbf{A}$ across the $\beta$ buckets. Similarly, the space used by the algorithm is $\tilde{\mathcal{O}}(Td)$. Once the gradient descent stage of Algorithm 2 begins, it takes $\text{poly}(d)$ time in each step $t \in [T]$ to query the $\beta = \Theta(Td)$ instances of SAMPLER and ESTIMATOR, for total time $T \cdot \text{poly}(d, \log n)$. □

## 3   SECOND-ORDER OPTIMIZATION

In this section, we repurpose our data stucture that performs importance sampling for SGD to instead perform importance sampling for second-order optimization. Given a second-order optimization algorithm that requires a sampled Hessian $\mathbf{H}_t$, possibly along with additional inputs such as the current iterate $\mathbf{x}_t$ and the gradient $\mathbf{g}_t$ of $F$, we model the update rule by an oracle $\mathbb{O}(\mathbf{H}_t)$, suppressing other inputs to the oracle in the notation. For example, the oracle $\mathbb{O}$ corresponding to the canonical second-order algorithm Newton's method can be formulated as

$$\mathbf{x}_{t+1} = \mathbb{O}(\mathbf{x}_t) := \mathbf{x}_t - [\mathbf{H}_t]^{-1}\mathbf{g}_t.$$

By black-boxing the update rule of any second-order optimization algorithm into the oracle, we can focus our attention to the running time of sampling a Hessian with nearly the optimal probability distribution. Thus we prove generalizations of the $L_2$ polynomial inner product sketch, the $L_2$ polynomial inner product sampler, and the leverage score sampler for Hessians.

**Theorem 3.1** *For a fixed $\epsilon > 0$ and polynomial $f$, there exists a data structure* HESTIMATOR *that outputs a $(1 + \epsilon)$-approximation to $\sum_{i=1}^n \left\| f(\langle \mathbf{a}_i, \mathbf{x} \rangle) \cdot \mathbf{a}_i^\top \mathbf{a}_i \right\|_F^2$ for any query $\mathbf{x} \in \mathbb{R}^d$. The data*

*structure requires a single pass over* $\mathbf{A} = \mathbf{a}_1 \circ \ldots \circ \mathbf{a}_n$ *(possibly through turnstile updates), can be built in* $\tilde{\mathcal{O}}\left(\text{nnz}(\mathbf{A}) + \frac{d}{\epsilon^2}\right)$ *time and* $\tilde{\mathcal{O}}\left(\frac{d}{\epsilon^2}\right)$ *space, uses query time* $\text{poly}\left(d, \frac{1}{\epsilon}, \log n\right)$, *and succeeds with probability* $1 - \frac{1}{\text{poly}(n)}$.

**Theorem 3.2** *For a fixed* $\epsilon > 0$ *and polynomial* $f$, *there exists a data structure* HSAMPLER *that takes any query* $\mathbf{x} \in \mathbb{R}^d$ *and outputs an index* $i \in [n]$ *with probability* $\frac{(1\pm\epsilon)\cdot\left\|f(\langle\mathbf{a}_i,\mathbf{x}\rangle)\cdot\mathbf{a}_i^\top\mathbf{a}_i\right\|_F^2}{\sum_{i=1}^{n}\left\|f(\langle\mathbf{a}_i,\mathbf{x}\rangle)\cdot\mathbf{a}_i^\top\mathbf{a}_i\right\|_F^2} + \frac{1}{\text{poly}(n)}$, *along with a matrix* $\mathbf{U} := f(\langle\mathbf{a}_i,\mathbf{x}\rangle)\cdot\mathbf{a}_i^\top\mathbf{a}_i + \mathbf{V}$, *where* $\mathbb{E}\left[\mathbf{V}\right] = 0$ *and* $\|\mathbf{V}\|_F \leq \epsilon \cdot \left\|f(\langle\mathbf{a}_i,\mathbf{x}\rangle)\cdot\mathbf{a}_i^\top\mathbf{a}_i\right\|_F$. *The data structure requires a single pass over* $\mathbf{A} = \mathbf{a}_1 \circ \ldots \circ \mathbf{a}_n$ *(possibly through turnstile updates), can be built in* $\tilde{\mathcal{O}}\left(\text{nnz}(\mathbf{A}) + \frac{d}{\epsilon^2}\right)$ *time and* $\tilde{\mathcal{O}}\left(\frac{d}{\epsilon^2}\right)$ *space, uses query time* $\text{poly}\left(d, \frac{1}{\epsilon}, \log n\right)$, *and succeeds with probability* $1 - \frac{1}{\text{poly}(n)}$.

**Theorem 3.3** *There exists an algorithm* HLEVERAGE *that returns all indices* $i \in [n]$ *such that* $\frac{(1\pm\epsilon)\cdot\left\|f(\langle\mathbf{a}_i,\mathbf{x}\rangle)\cdot\mathbf{a}_i^\top\mathbf{a}_i\right\|_F^2}{\sum_{i=1}^{n}\left\|f(\langle\mathbf{a}_i,\mathbf{x}\rangle)\cdot\mathbf{a}_i^\top\mathbf{a}_i\right\|_F^2} \geq \frac{1}{200Td}$ *for some* $\mathbf{x} \in \mathbb{R}^n$, *along with a matrix* $\mathbf{U}_i := f(\langle\mathbf{a}_i,\mathbf{x}\rangle)\cdot\mathbf{a}_i^\top\mathbf{a}_i + \mathbf{V}_i$, *where* $\|\mathbf{V}_i\|_F \leq \epsilon \cdot \left\|f(\langle\mathbf{a}_i,\mathbf{x}\rangle)\cdot\mathbf{a}_i^\top\mathbf{a}_i\right\|_F$. *The algorithm uses requires a single pass over* $\mathbf{A} = \mathbf{a}_1 \circ \ldots \circ \mathbf{a}_n$ *(possibly through turnstile updates), uses* $\tilde{\mathcal{O}}\left(\text{nnz}(\mathbf{A}) + \frac{d^\omega}{\epsilon^2}\right)$ *runtime (where* $\omega$ *denotes the exponent of square matrix multiplication) and* $\tilde{\mathcal{O}}\left(\frac{d}{\epsilon^2}\right)$ *space, and succeeds with probability* $1 - \frac{1}{\text{poly}(n)}$.

We remark that HSAMPLER and LEVERAGE are generalizations of ESTIMATOR and SAMPLER that simply return an outer product of a noisy vector rather than the noisy vector itself.

As before, observe that we could simply run an instance of HSAMPLER to sample a Hessian through importance sampling, but sampling $T$ Hessians requires $T$ independent instances, significantly increasing the total runtime. We thus use the same two level data structure that partitions the rows of matrix $\mathbf{A} = \mathbf{a}_1 \circ \ldots \circ \mathbf{a}_n$ into $\beta := \Theta(Td)$ buckets $B_1, \ldots, B_\beta$. We then create an instance of HESTIMATOR and HSAMPLER for each bucket. For an iterate $\mathbf{x}_t$, we sample $j \in [\beta]$ among the buckets $B_1, \ldots, B_\beta$ with probability roughly $\frac{\sum_{i \in B_j}\left\|f(\langle\mathbf{a}_i,\mathbf{x}_t\rangle)\cdot\mathbf{a}_i^\top\mathbf{a}_i\right\|_F^2}{\sum_{i=1}^{n}\left\|f(\langle\mathbf{a}_i,\mathbf{x}_t\rangle)\cdot\mathbf{a}_i^\top\mathbf{a}_i\right\|_F^2}$ using HESTIMATOR and then querying HSAMPLER$_j$ at $\mathbf{x}_t$ to sample a Hessian among the indices partitioned into bucket $B_j$. As before, this argument fails when the same bucket $B_j$ is sampled multiple times, due to dependencies in randomness, but this issue can be avoided by using HLEVERAGE to decrease the probability that each bucket is sampled. We give the algorithm in full in Algorithm 3.

We remark that Algorithm 3 can be generalized to handle oracles $\mathbb{O}$ corresponding to second-order methods that require batches of subsampled Hessians in each iteration. For example, if we want to run $T$ iterations of a second-order method that requires $s$ subsampled Hessians in each batch, we can simply modify Algorithm 3 to sample $s$ Hessians in each iteration as input to $\mathbb{O}$ and thus $Ts$ Hessians in total.

## 4 EMPIRICAL EVALUATIONS

Our primary contribution is the theoretical design of a nearly input sparsity time algorithm that approximates optimal importance sampling SGD. In this section we implement a scaled-down version of our algorithm and compare its performance on large-scale real world datasets to SGD with uniform sampling on logistic regression. We also consider both linear regression and support-vector machines (SVMs) in the supplementary material. Because most rows have roughly uniformly small leverage scores in real-world data, we assume that no bucket contains a row with a significantly large leverage score and thus the implementation of our importance sampling algorithm does not create multiple samplers for any buckets. By similar reasoning, our implementation uniformly samples a number of indices $i$ and estimates $\sum_{i=1}^{n}\left\|f(\langle\mathbf{a}_i,\mathbf{x}\rangle)\cdot\mathbf{a}_i\right\|_2^2$ by rescaling. Observe that although these simplifications to our algorithm decreases the wall-clock running time and the total space used by our algorithm, they only decrease the quality of our solution for each SGD iteration. We also consider two hybrid SGD sampling algorithms; the first takes the better gradient obtained at each iteration from both uniform sampling and importance sampling while the second performs 25 iterations of

---

**Algorithm 3** Second-Order Optimization with Importance Sampling

---

**Input:** Matrix $\mathbf{A} = \mathbf{a}_1 \circ \ldots \circ \mathbf{a}_n \in \mathbb{R}^{n \times d}$, parameter $T$ for number of sampled Hessians, oracle $\mathbb{O}$ that performs the update rule.

**Output:** $T$ approximate Hessians.

1: **Preprocessing Stage:**
2: $\beta \leftarrow \Theta(Td)$ with a sufficiently large constant.
3: Let $h : [n] \rightarrow [\beta]$ be a uniformly random hash function.
4: Let $\mathbf{B}_j$ be the matrix formed by the rows $\mathbf{a}_i$ of $\mathbf{A}$ with $h(i) = j$, for each $j \in [\beta]$.
5: Create an instance HESTIMATOR$_j$ and HSAMPLER$_j$ for each $\mathbf{B}_j$ with $j \in [\beta]$ with $\epsilon = \frac{1}{2}$.
6: Run HLEVERAGE to find a set $L_0$ of row indices and corresponding (noisy) outer products.
7: **Second-Order Optimization Stage:**
8: Randomly pick starting location $\mathbf{x}_0$
9: **for** $t = 1$ to $T$ **do**
10:    Let $q_i$ be the output of HESTIMATOR$_j$ on query $\mathbf{x}_{t-1}$ for each $i \in [\beta]$.
11:    Sample $j \in [\beta]$ with probability $p_j = \frac{q_j}{\sum_{i \in [\beta]} q_i}$.
12:    **if** there exists $i \in L_0$ with $h(i) = j$ **then**
13:        Use HESTIMATOR$_j$, HLEVERAGE, and HSAMPLER$_j$ to sample Hessian $\mathbf{H}_t$.
14:    **else**
15:        Use HSAMPLER$_j$ to sample Hessian $\mathbf{H}_t = \widehat{\nabla f_{i_t}(\mathbf{x}_t)}$.
16:    $\widehat{p_{i,t}} \leftarrow \frac{\|\mathbf{H}_t\|_F^2}{\sum_{j \in [\beta]} q_j}$
17:    $\mathbf{x}_{t+1} \leftarrow \mathbb{O}\left(\frac{1}{n\widehat{p_{i,t}}}\mathbf{H}_t\right)$

---

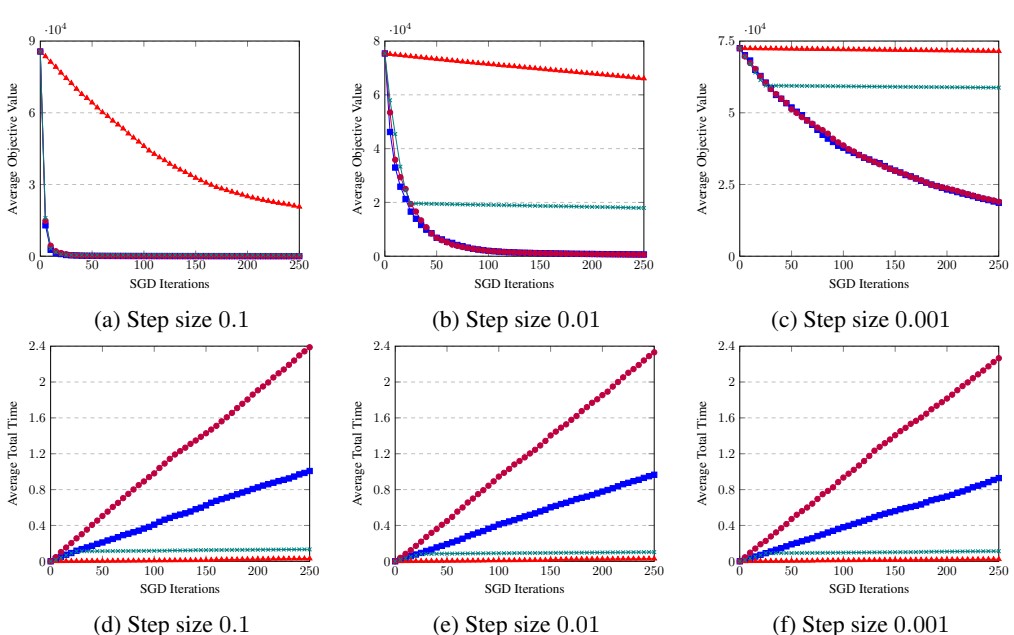

(a) Step size 0.1          (b) Step size 0.01          (c) Step size 0.001

(d) Step size 0.1          (e) Step size 0.01          (f) Step size 0.001

Fig. 1: Comparison of objective values and runtimes for importance sampling (in blue squares), uniform sampling (in red triangles), hybrid sampling that chooses the better gradient at each step (in purple circles), and hybrid sampling that performs 25 steps of importance sampling followed by uniform sampling (in teal X's) over various step-sizes for logistic regression on a9a Adult dataset from UCI, across 250 iterations, averaged over 10 repetitions.

importance sampling before using uniform sampling for the remaining iterations. Surprisingly, our SGD importance sampling implementation not only significantly improves upon SGD with uniform sampling, but are also competitive with the two hybrid algorithms. We do not consider other SGD variants due to either their distributional assumptions or lack of known flexibility to big data models. The experiments were performed in Python 3.6.9 on an Intel Core i7-8700K 3.70 GHz CPU with

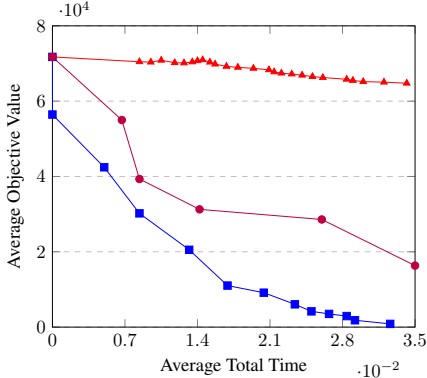

Fig. 2: Comparison of objective values and wall-clock time for importance sampling (in blue squares), uniform sampling (in red triangles), and hybrid sampling that chooses the better gradient at each step (in purple circles) over step-size 0.1 for logistic regression on a9a Adult dataset from UCI, averaged over 3 repetitions across approximately 15 minute total computation time.

12 cores and 64GB DDR4 memory, using a Nvidia Geforce GTX 1080 Ti 11GB GPU. Our code is publicly available at `https://github.com/SGD-adaptive-importance/code`.

**Logistic Regression.** We performed logistic regression on the a9a Adult data set collected by UCI and retrieved from LibSVM (Chang & Lin, 2011). The features correspond to responses from the 1994 Census database and the prediction task is to determine whether a person makes over 50K USD a year. We trained using a data batch of 32581 points and 123 features and tested the performance on a separate batch of 16281 data points. For each evaluation, we generated 10 random initial positions shared for importance sampling and uniform sampling. We then ran 250 iterations of SGD for each of the four algorithms, creating only 250 buckets for the importance sampling algorithm and computed the average performance on each iteration across these 10 separate instances. The relative average performance of all algorithms was relatively robust to the step-size. Although uniform sampling used significantly less time overall, our importance sampling SGD algorithm actually had better performance when considering either number of iterations or wall-clock time across all tested step-sizes. For example, uniform sampling had average objective value 20680 at iteration 250 using 0.0307 seconds with step-size 0.1, but importance sampling had average objective value 12917 at iteration 5 using 0.025 seconds. We give our results for logistic regression in Figure 1. We repeat our experiments in Figure 2 to explicitly compare the objective value of each algorithm with respect to wall-clock time, rather than SGD iterations. Thus our results in Figure 2 empirically demonstrate the advantages of our algorithm across the most natural metrics. For additional experiments, see Section B.

## 5 CONCLUSION AND FUTURE WORK

We have given variance reduction methods for both first-order and second-order stochastic optimization. Our algorithms require a single pass over the data, which may even arrive implicitly in the form of turnstile updates, and use input sparsity time and $\tilde{\mathcal{O}}(Td)$ space. Our algorithms are also amenable to big data models such as the streaming and distributed models and are supported by empirical evaluations on large-scale datasets. We believe there are many interesting future directions to explore. For example, can we generalize our techniques to show provable guarantees for other SGD variants and accelerated methods? A very large-scale empirical study of these methods would also be quite interesting.

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

## A    DISCUSSION, FULL ALGORITHMS, AND PROOFS

For the sake of presentation, we consider the case where $p = 2$; higher dimensions follow from the same approach, using tensor representation instead of matrix representation. Instead of viewing the input matrix $\mathbf{A} = \mathbf{a}_1 \circ \ldots \circ \mathbf{a}_n \in \mathbb{R}^{n \times d}$ as a number of rows, we instead view the matrix $\mathbf{A} = \mathbf{A}_1 \circ \ldots \circ \mathbf{A}_n \in \mathbb{R}^{nd \times d}$, where each matrix $\mathbf{A}_i = \mathbf{a}_i \otimes \mathbf{a}_i$ is the outer product of the row $\mathbf{a}_i$ with itself.

### A.1    $L_2$ POLYNOMIAL INNER PRODUCT SAMPLER

For ease of discussion, we describe in this section a data structure that allows sampling an index $i \in [n]$ with probability approximately $\frac{\|\mathbf{A}_i \mathbf{x}\|_{1,2,d}}{\|\mathbf{A} \mathbf{x}\|_{1,2,d}}$ in linear time and sublinear space, where for a matrix $\mathbf{A} \in \mathbb{R}^{nd \times d}$, we use $\|\mathbf{A} \mathbf{x}\|_{1,2,d}$ to denote $\sum_{i=1}^{n} \|\mathbf{A}_i \mathbf{x}\|_2$, where each $\mathbf{A}_i \in \mathbb{R}^{d \times d}$ and $\mathbf{A} = \mathbf{A}_1 \circ \ldots \circ \mathbf{A}_n$. The generalization to a $L_2$ polynomial inner product sampler follows immediately. Notably, our data structure can be built simply given access to $\mathbf{A}$, and will still sample from the correct distribution when $\mathbf{x}$ is given as a post-processing vector. We first describe in Section A.1.1 some necessary subroutines that our sampler requires. These subroutines are natural generalizations of the well-known frequency moment estimation algorithm of Alon et al. (1999) and heavy hitter detection algorithm of Charikar et al. (2004). We then give the $L_{1,2,d}$ sampler in full in Section A.1.2.

#### A.1.1    FREQUENCY MOMENT AND HEAVY HITTER GENERALIZATIONS

We first recall a generalization to the frequency moment estimation algorithm by Alon et al. (1999) that also supports post-processing multiplication by any vector $\mathbf{x} \in \mathbb{R}^d$.

**Lemma A.1** *(Mahabadi et al., 2020) Given a constant $\epsilon > 0$, there exists a one-pass streaming algorithm* AMS *that takes updates to entries of a matrix $\mathbf{A} \in \mathbb{R}^{n \times d}$, as well as query access to post-processing vectors $\mathbf{x} \in \mathbb{R}^d$ and $\mathbf{v} \in \mathbb{R}^d$ that arrive after the stream, and outputs a quantity $\hat{F}$ such that $(1 - \epsilon) \|\mathbf{A}\mathbf{x} - \mathbf{v}\|_2 \leq \hat{F} \leq (1 + \epsilon) \|\mathbf{A}\mathbf{x} - \mathbf{v}\|_2$. The algorithm uses $\mathcal{O}\left(\frac{d}{\epsilon^2} \left(\log^2 n + \log \frac{1}{\delta}\right)\right)$ bits of space and succeeds with probability at least $1 - \delta$.*

---

**Algorithm 4** Basic algorithm COUNTSKETCH that outputs heavy submatrices of $\|\mathbf{A}\mathbf{x}\|_{1,2,d}$, where $\mathbf{x}$ is a post-processing vector

---

**Input:** Matrix $\mathbf{A} \in \mathbb{R}^{nd \times d}$, post-processing vector $\mathbf{x} \in \mathbb{R}^d$, constant parameter $\epsilon > 0$.
**Output:** Slight perturbations of the vector $\mathbf{A}_i \mathbf{x}$ for which $\|\mathbf{A}_i \mathbf{x}\|_2 \geq \epsilon \|\mathbf{A}\mathbf{x}\|_{1,2,d}$.
 1: $r \leftarrow \Theta(\log n)$ with a sufficiently large constant.
 2: $b \leftarrow \Omega\left(\frac{1}{\epsilon^2}\right)$ with a sufficiently large constant.
 3: Let $\mathcal{T}$ be an $r \times b$ table of buckets, where each bucket stores an $\mathbb{R}^{d \times d}$ matrix, initialized to the zeros matrix.
 4: Let $s_i \in \{-1, +1\}$ be 4-wise independent for $i \in [n]$.
 5: Let $h_i : [n] \to [b]$ be 4-wise independent for $i \in [r]$.
 6: **Process A:**
 7: Let $\mathbf{A} = \mathbf{A}_1 \circ \ldots \circ \mathbf{A}_n$, where each $\mathbf{A}_i \in \mathbb{R}^{d \times d}$.
 8: **for** each $j = 1$ to $n$ **do**
 9:     **for** each $i = 1$ to $r$ **do**
10:         Add $s_j \mathbf{A}_j$ to the matrix in bucket $h_i(j)$ of row $i$.
11: Let $\mathbf{M}_{i,j}$ be the matrix in row $i$, bucket $j$ of $\mathcal{T}$ for $i \in [r], j \in [b]$.
12: **Process x:**
13: **for** $i \in [r], j \in [b]$ **do**
14:     $\mathbf{M}_{i,j} \leftarrow \mathbf{M}_{i,j} \mathbf{x}$
15: On query $k \in [n]$, report $\text{median}_{i \in [r]} \left\|\mathbf{M}_{i, h_i(k)}\right\|_2$.

---

Let $\mathbf{A}_1, \ldots, \mathbf{A}_n \in \mathbb{R}^{d \times d}$ and $\mathbf{A} = \mathbf{A}_1 \circ \ldots \circ \mathbf{A}_n \in \mathbb{R}^{nd \times d}$. Let $\mathbf{x} \in \mathbb{R}^{d \times 1}$ be a post-processing vector that is revealed only after $\mathbf{A}$ has been completely processed. For a given $\epsilon > 0$, we say a block $\mathbf{A}_i$ with $i \in [n]$ is *heavy* if $\|\mathbf{A}_i \mathbf{x}\|_2 \geq \epsilon \|\mathbf{A}\mathbf{x}\|_{1,2,d}$. We show in Algorithm 4 an algorithm that

processes $\mathbf{A}$ into a sublinear space data structure and identifies the heavy blocks of $\mathbf{A}$ once $\mathbf{x}$ is given. Moreover, for each heavy block $\mathbf{A}_i$, the algorithm outputs a vector $\mathbf{y}$ that is a good approximation to $\mathbf{A}_i\mathbf{x}$. The algorithm is a natural generalization of the CountSketch heavy-hitter algorithm introduced by Charikar et al. (2004).

For a vector $\mathbf{v} \in \mathbb{R}^{nd \times 1}$, we use $\mathbf{v}_{tail(b)}$ to denote $\mathbf{v}$ with the $b$ blocks of $d$ rows of $\mathbf{v}$ with the largest $\ell_2$ norm set to zeros.

**Lemma A.2** *There exists an algorithm that uses* $\mathcal{O}\left(\frac{1}{\epsilon^2}d^2\left(\log^2 n + \log\frac{1}{\delta}\right)\right)$ *space that outputs a vector* $\mathbf{y}_i$ *for each index* $i \in [n]$ *so that* $\left| \|\mathbf{y}_i\|_2 - \|\mathbf{A}_i\mathbf{x}\|_2 \right| \leq \epsilon\left\|(\mathbf{Ax})_{tail\left(\frac{2}{\epsilon^2}\right)}\right\|_2 \leq \epsilon\left\|(\mathbf{Ax})_{tail\left(\frac{2}{\epsilon^2}\right)}\right\|_{1,2,d}$ *with probability at least* $1 - \delta$. *Moreover if* $\mathbf{Y} = \mathbf{y}_1 \circ \ldots \circ \mathbf{y}_n$, *then* $\left\|(\mathbf{Ax})_{tail\left(\frac{2}{\epsilon^2}\right)}\right\|_2 \leq \left\|\mathbf{Ax} - \widehat{\mathbf{Y}}\right\|_2 \leq 2\left\|(\mathbf{Ax})_{tail\left(\frac{2}{\epsilon^2}\right)}\right\|_2$ *with probability at least* $1 - \delta$, *where* $\widehat{\mathbf{Y}} = \mathbf{Y} - \mathbf{Y}_{tail\left(\frac{2}{\epsilon^2}\right)}$ *denotes the top* $\frac{2}{\epsilon^2}$ *blocks of* $\mathbf{Y}$ *by* $\ell_2$ *norm.*

**Proof :** Fix an index $i \in [n]$. Consider the estimate of $\|\mathbf{A}_i\mathbf{x}\|_2$ in row $\alpha$ of the CountSketch table $\mathcal{T}$. Then $h_\alpha(i)$ is the bucket of $\mathcal{T}$ in row $\alpha$ to which $\mathbf{A}_i\mathbf{x}$ hashes. Let $\mathcal{E}_1$ be the event that the $\frac{2}{\epsilon^2}$ blocks of size $d$ of $\mathbf{Ax}$ with the largest $\ell_2$ norm are not hashed to $h_\alpha(i)$. Observe that for $b = \Omega\left(\frac{1}{\epsilon^2}\right)$ with sufficiently large constant, $\mathcal{E}_1$ occurs with probability at least $\frac{1}{12}$ by a union bound.

Let $\mathbf{v}$ be the sum of the vectors representing the blocks that are hashed to bucket $h_\alpha(i)$ excluding $\mathbf{A}_i\mathbf{x}$, so that $\mathbf{v}$ is the noise for the estimate of $\mathbf{A}_i\mathbf{x}$ in row $\alpha$. Conditioned on $\mathcal{E}_1$, we can bound the expected squared norm of the noise in bucket $h_\alpha(i)$ for sufficiently large $b$ by $\mathbb{E}\left[\|\mathbf{v}\|_2^2\right] \leq \frac{\epsilon^2}{9}\left\|(\mathbf{Ax})_{tail\left(\frac{2}{\epsilon^2}\right)}\right\|_2^2$. Hence we have $\mathrm{Var}(\|\mathbf{v}_i\|_2) \leq \frac{\epsilon^2}{9}\left\|(\mathbf{Ax})_{tail\left(\frac{2}{\epsilon^2}\right)}\right\|_2^2$. Thus from Jensen's inequality, Chebyshev's inequality and conditioning on $\mathcal{E}_1$,

$$\mathbf{Pr}\left[\|\mathbf{v}_i\|_2 \geq \epsilon\left\|(\mathbf{Ax})_{tail\left(\frac{2}{\epsilon^2}\right)}\right\|_2\right] \leq \frac{1}{4} + \frac{1}{12} = \frac{2}{3}.$$

The first claim then follows from the observation that $\left\|(\mathbf{Ax})_{tail\left(\frac{2}{\epsilon^2}\right)}\right\|_2 \leq \left\|(\mathbf{Ax})_{tail\left(\frac{2}{\epsilon^2}\right)}\right\|_{1,2,d}$ and noting that we can boost the probability of success to $1 - \frac{1}{\mathrm{poly}(n)}$ by repeating for each of the $r = \Theta(\log n)$ rows and taking the median.

Finally, observe that $\left\|(\mathbf{Ax})_{tail\left(\frac{2}{\epsilon^2}\right)}\right\|_2 \leq \left\|\mathbf{Ax} - \widehat{\mathbf{Y}}\right\|_2$, since $\widehat{\mathbf{Y}}$ has at most $\frac{2}{\epsilon^2}$ nonzero blocks, while $(\mathbf{Ax})_{tail\left(\frac{2}{\epsilon^2}\right)}$ has all zeros in the $\frac{2}{\epsilon^2}$ blocks of $\mathbf{Ax}$ with the largest $\ell_2$ norm. Since $\mathbf{Ax} - \widehat{\mathbf{Y}}$ alters at most $\frac{2}{\epsilon^2}$ rows of $\mathbf{Ax}$, each by at most $\epsilon\left\|(\mathbf{Ax})_{tail\left(\frac{2}{\epsilon^2}\right)}\right\|_2$, then

$$\left\|\mathbf{Ax} - \widehat{\mathbf{Y}}\right\|_2 \leq \sqrt{\sum_{i=1}^{2/\epsilon^2}\left(\epsilon\left\|(\mathbf{Ax})_{tail\left(\frac{2}{\epsilon^2}\right)}\right\|_2\right)^2} = 2\left\|(\mathbf{Ax})_{tail\left(\frac{2}{\epsilon^2}\right)}\right\|_2.$$

$\square$

### A.1.2 SAMPLING ALGORITHM

Our approach is similar to $\ell_p$ sampling techniques in Andoni et al. (2011); Jowhari et al. (2011), who consider sampling indices in vectors, and almost verbatim to Mahabadi et al. (2020), who consider sampling rows in matrices given post-processing multiplication by a vector.

The high level idea is to note that if $t_i \in [0, 1]$ is chosen uniformly at random, then

$$\mathbf{Pr}\left[\frac{\|\mathbf{A}_i\|_{1,2,d}}{t_i} \geq \|\mathbf{A}\|_{1,2,d}\right] = \frac{\|\mathbf{A}_i\|_{1,2,d}}{\|\mathbf{A}\|_{1,2,d}}.$$

Thus if $\mathbf{B}_i = \frac{\mathbf{A}_i}{t_i}$ and there exists exactly one index $i$ such that $\|\mathbf{B}_i\|_{1,2,d} \geq \|\mathbf{A}\|_{1,2,d}$, then the task would reduce to outputting $B_j$ that maximizes $\|\mathbf{B}_j\|_{1,2,d}$ over all $j \in [n]$. In fact, we can show that

$\mathbf{B}_i$ is an $\mathcal{O}(\epsilon)$-heavy hitter of $\mathbf{B}$ with respect to the $L_{1,2,d}$ norm. Hence, we use a generalization of COUNTSKETCH to identify the heavy hitters of $\mathbf{B}$, approximate the maximum index $i$, and check whether $\|\mathbf{B}_i\|_{1,2,d}$ is at least (an estimate of) $\|\mathbf{A}\|_{1,2,d}$.

Unfortunately, this argument might fail due to several reasons. Firstly, there might exists zero or multiple indices $i$ such that $\|\mathbf{B}_i\|_{1,2,d} \geq \|\mathbf{A}\|_{1,2,d}$. Then the probability distribution that an index $i$ satisfies $\|\mathbf{B}_i\|_{1,2,d} \geq \|\mathbf{A}\|_{1,2,d}$ and that $\|\mathbf{B}_i\|_{1,2,d} > \|\mathbf{B}_j\|_{1,2,d}$ for all other $j \in [n]$ is not the same as the desired distribution. Fortunately, we show that this only happens with small probability, slightly perturbing the probability of returning each $i \in [n]$.

Another possibility is that the error in COUNTSKETCH is large enough to misidentify whether $\|\mathbf{B}_i\|_{1,2,d} \geq \|\mathbf{A}\|_{1,2,d}$. Using a statistical test, this case can usually be identified and so the algorithm will be prevented from outputting a sample in this case. Crucially, the probability that the algorithm is aborted by the statistical test is roughly independent of which index achieves the maximum. As a result, the probability of returning each $i \in [n]$ is within a $(1 \pm \epsilon)$ factor of $\frac{\|\mathbf{A}_i\|_{1,2,d}}{\|\mathbf{A}\|_{1,2,d}}$ when the algorithm does not abort.

We show that the probability that algorithm succeeds is $\Theta(\epsilon)$ so then running $\mathcal{O}\left(\log \frac{1}{\epsilon}\right)$ instances of the algorithm suffices to output some index from the desired distribution with constant probability, or abort otherwise. Because the underlying data structure is a linear sketch, then it is also robust to post-processing multiplication by any vector $\mathbf{x} \in \mathbb{R}^d$. Finally, we note that although our presentation refers to the scaling factors $t_i$ as independent random variables, our analysis shows that they only need to be $\mathcal{O}(1)$-wise independent and thus we can generate the scaling factors in small space in the streaming model. We give the $L_{1,2,d}$ sampler in Algorithm 5.

---

**Algorithm 5** $L_{1,2,d}$ Sampler

---

**Input:** Matrix $\mathbf{A} \in \mathbb{R}^{nd \times d}$ with $\mathbf{A} = \mathbf{A}_1 \circ \ldots \circ \mathbf{A}_n$, where each $\mathbf{A}_i \in \mathbb{R}^{d \times d}$, vector $\mathbf{x} \in \mathbb{R}^{d \times 1}$ that arrives after processing $\mathbf{A}$, constant parameter $\epsilon > 0$.
**Output:** Noisy $\mathbf{A}_i \mathbf{x}$ of $\mathbf{A}\mathbf{x}$ sampled roughly proportional to $\|\mathbf{A}_i \mathbf{x}\|_2$.
 1: **Pre-processing Stage:**
 2: $b \leftarrow \Omega\left(\frac{1}{\epsilon^2}\right)$, $r \leftarrow \Theta(\log n)$ with sufficiently large constants
 3: For $i \in [n]$, generate independent scaling factors $t_i \in [0,1]$ uniformly at random.
 4: Let $\mathbf{B}$ be the matrix consisting of matrices $\mathbf{B}_i = \frac{1}{t_i}\mathbf{A}_i$.
 5: Let ESTIMATOR and AMS track the $L_{1,2,d}$ norm of $\mathbf{A}\mathbf{x}$ and Frobenius norm of $\mathbf{B}\mathbf{x}$, respectively.
 6: Let COUNTSKETCH be an $r \times b$ table, where each entry is a matrix in $\mathbb{R}^{d \times d}$.
 7: **for** each submatrix $\mathbf{A}_i$ **do**
 8:                                                                                  ▷Process $\mathbf{A}$:
 9:      Update COUNTSKETCH with $\mathbf{B}_i = \frac{1}{t_i}\mathbf{A}_i$.
10:      Update linear sketch ESTIMATOR with $\mathbf{A}_i$.
11:      Update linear sketch AMS with $\mathbf{B}_i = \frac{1}{t_i}\mathbf{A}_i$.
12: Post-process $\mathbf{x}$ in AMS, COUNTSKETCH, and ESTIMATOR.               ▷Process $\mathbf{x}$:
13: **Sample a submatrix:**
14: Use ESTIMATOR to compute $\widehat{F}$ with $\|\mathbf{A}\mathbf{x}\|_{1,2,d} \leq \widehat{F} \leq 2\|\mathbf{A}\mathbf{x}\|_{1,2,d}$.
15: Extract the $\frac{2}{\epsilon^2}$ (noisy) blocks of $d$ rows of $\mathbf{B}\mathbf{x}$ with the largest estimated $\ell_2$ norms by COUNTSKETCH.
16: Let $\mathbf{M} \in \mathbb{R}^{nd \times 1}$ be the $\frac{2}{\epsilon^2}$-block sparse matrix consisting of these top (noisy) block.
17: Use AMS to compute $\widehat{S}$ with $\|\mathbf{B}\mathbf{x} - \mathbf{M}\|_2 \leq \widehat{S} \leq 2\|\mathbf{B}\mathbf{x} - \mathbf{M}\|_2$.
18: Let $\mathbf{r}_i$ be the (noisy) block of $d$ rows in COUNTSKETCH with the largest norm.
19: **if** $\widehat{S} > \widehat{F}\sqrt{\log \frac{1}{\epsilon}}$ or $\|\mathbf{r}_i\|_2 < \frac{1}{\epsilon}\widehat{F}$ **then**
20:      **Return** FAIL.
21: **else**
22:      **Return** $\mathbf{r} = t_i \mathbf{r}_i$.

---

We first show that the probability that Algorithm 5 returns FAIL is independent of which index $i \in [n]$ achieves $\text{argmax}_{i \in [n]} \frac{1}{t_i} \|\mathbf{A}_i \mathbf{x}\|_2$.

**Lemma A.3** *Let $i \in [n]$ and fix a value of $t_i \in [0, 1]$ uniformly at random. Then conditioned on the value of $t_i$,*

$$\mathbf{Pr}\left[\widehat{S} > \widehat{F}\sqrt{\log\frac{1}{\epsilon}}\right] = \mathcal{O}\left(\epsilon\right) + \frac{1}{\text{poly}(n)}.$$

**Proof :** We first observe that if we upper bound $\widehat{S}$ by $4\left\|(\mathbf{Bx})_{tail\left(\frac{2}{\epsilon^2}\right)}\right\|_2$ and lower bound $\widehat{F}$ by $\|\mathbf{Ax}\|_{1,2,d}$, then it suffices to show that the probability of $4\left\|(\mathbf{Bx})_{tail\left(\frac{2}{\epsilon^2}\right)}\right\|_2 > \sqrt{\log\frac{1}{\epsilon}}\,\|\mathbf{Ax}\|_{1,2,d}$ is small. Thus we define $\mathcal{E}_1$ as the event that:

(1) $\|\mathbf{Ax}\|_{1,2,d} \leq \widehat{F} \leq 2\|\mathbf{Ax}\|_{1,2,d}$

(2) $\|\mathbf{Bx} - \mathbf{M}\|_F \leq \widehat{S} \leq 2\|\mathbf{Bx} - \mathbf{M}\|_F$

(3) $\left\|(\mathbf{Bx})_{tail\left(\frac{2}{\epsilon^2}\right)}\right\|_2 \leq \|\mathbf{Bx} - \mathbf{M}\|_F \leq 2\left\|(\mathbf{Bx})_{tail\left(\frac{2}{\epsilon^2}\right)}\right\|_2$

Note that by Theorem 2.1, Lemma A.2 and Lemma A.1, $\mathcal{E}_1$ holds with high probability.

Let $U = \|\mathbf{Ax}\|_{1,2,d}$. For each block $\mathbf{A}_j\mathbf{x}$, we define $y_j$ to be the indicator variable for whether the scaled block $\mathbf{B}_j\mathbf{x}$ is heavy, so that $y_j = 1$ if $\|\mathbf{B}_j\mathbf{x}\|_2 > U$ and $y_j = 0$ otherwise. We also define $z_j \in [0, 1]$ as a scaled random variable for whether $\mathbf{B}_j\mathbf{x}$ is light and how much squared mass it contributes, $z_j = \frac{1}{U^2}\|\mathbf{B}_j\mathbf{x}\|_2^2 (1 - y_j)$. Let $Y = \sum_{j \neq i} y_j$ be the total number of heavy blocks besides $\mathbf{B}_i\mathbf{x}$ and $Z = \sum_{j \neq i} z_j$ be the total scaled squared mass of the small rows. Let $\mathbf{h} \in \mathbb{R}^{nd}$ be the vector that contains the heavy blocks so that coordinates $(j-1)d + 1$ through $jd$ of $\mathbf{h}$ correspond to $\mathbf{B}_j\mathbf{x}$ if $y_j = 1$ and they are all zeros otherwise. Hence, $\mathbf{h}$ contains at most $Y + 1$ nonzero blocks and thus at most $(Y + 1)d$ nonzero entries. Moreover, $U^2 Z = \|\mathbf{Bx} - \mathbf{h}\|_2^2$ and $\left\|(\mathbf{Bx})_{tail\left(\frac{2}{\epsilon^2}\right)}\right\|_2 \leq U\sqrt{Z}$ unless $Y \geq \frac{2}{\epsilon^2}$.

Thus if we define $\mathcal{E}_2$ to be the event that $Y \geq \frac{2}{\epsilon^2}$ and $\mathcal{E}_3$ to be the event that $Z \geq \frac{1}{16U^2}\log\frac{1}{\epsilon}\,\|\mathbf{Ax}\|_{1,2,d}^2$, then $\neg\mathcal{E}_2 \wedge \neg\mathcal{E}_3$ implies $4\left\|(\mathbf{Bx})_{tail\left(\frac{2}{\epsilon^2}\right)}\right\|_2 \leq \sqrt{\log\frac{1}{\epsilon}}\,\|\mathbf{Ax}\|_{1,2,d}$, so it suffices to bound the probability of the events $\mathcal{E}_2$ and $\mathcal{E}_3$ by $\mathcal{O}\left(\epsilon\right)$. Intuitively, if the number of heavy rows is small ($\neg\mathcal{E}_2$) and the total contribution of the small rows is small ($\neg\mathcal{E}_3$), then the tail estimator is small, so the probability of failure due to the tail estimator is small.

To analyze $\mathcal{E}_2$, note that $y_j = 1$ if and only if $\frac{1}{t_j}\|\mathbf{A}_j\mathbf{x}\|_2 > U$, so $\mathbb{E}\left[y_i\right] = \frac{\|\mathbf{A}_j\mathbf{x}\|_2}{U}$ and thus $\mathbb{E}\left[Y\right] \leq 1$ since $Y = \sum_{j \neq i} y_j$ and $U = \|\mathbf{Ax}\|_{1,2,d} = \sum_j \|\mathbf{A}_j\mathbf{x}\|_2$. We also have $\text{Var}(Y) \leq 1$ so that $\mathbf{Pr}\left[\mathcal{E}_2\right] = \mathcal{O}\left(\epsilon\right)$ for sufficiently small $\epsilon$, by Chebyshev's inequality.

To analyze $\mathcal{E}_3$, recall that $z_j = \frac{1}{U^2}\|\mathbf{B}_j\mathbf{x}\|_2^2 (1 - y_j)$. Thus $z_j > 0$ only if $y_j = 0$ or equivalently, $\|\mathbf{B}_j\mathbf{x}\|_2 \leq U$. Since $\mathbf{B}_j\mathbf{x} = \frac{1}{t_j}\mathbf{A}_j\mathbf{x}$, then $z_j > 0$ only if $t_j \geq \frac{\|\mathbf{A}_j\mathbf{x}\|_2}{\|\mathbf{Ax}\|_{1,2,d}}$. Therefore,

$$\mathbb{E}\left[z_j\right] \leq \int_{\|\mathbf{A}_j\mathbf{x}\|_2/\|\mathbf{Ax}\|_{1,2,d}}^{\infty} z_j\, dt_j = \int_{\|\mathbf{A}_j\mathbf{x}\|_2/\|\mathbf{Ax}\|_{1,2,d}}^{\infty} \frac{1}{t_j^2}\frac{1}{U^2}\|\mathbf{A}_j\mathbf{x}\|_2^2\, dt_j \leq \frac{\|\mathbf{A}_j\mathbf{x}\|_2}{\|\mathbf{Ax}\|_{1,2,d}}.$$

Since $Z = \sum_{j \neq i} z_j$, then $\mathbb{E}\left[Z\right] \leq 1$ and similarly $\text{Var}(Z) \leq 1$. Hence by Bernstein's inequality, $\mathbf{Pr}\left[Z > \frac{1}{16}\log\frac{1}{\epsilon}\right] = \mathcal{O}\left(\epsilon\right)$, so then $\mathbf{Pr}\left[\mathcal{E}_3\right] = \mathcal{O}\left(\epsilon\right)$. Thus $\mathbf{Pr}\left[\neg\mathcal{E}_1 \vee \mathcal{E}_2 \vee \mathcal{E}_3\right] = \mathcal{O}\left(\epsilon\right) + \frac{1}{\text{poly}(n)}$, as desired. $\square$

We now show that Algorithm 5 outputs a noisy approximation to $\mathbf{A}_i\mathbf{x}$, where $i \in [n]$ is drawn from approximately the correct distribution, i.e., the probability of failure does not correlate with the index that achieves the maximum value.

**Lemma A.4** *For a fixed value of $\widehat{F}$, the probability that Algorithm 5 outputs (noisy) submatrix $\mathbf{A}_i\mathbf{x}$ is $(1 \pm \mathcal{O}\left(\epsilon\right))\frac{\|\mathbf{A}_i\mathbf{x}\|_2}{\widehat{F}} + \frac{1}{\text{poly}(n)}$.*

**Proof :** Let $\mathcal{E}$ be the event that $t_i < \frac{\epsilon \|\mathbf{A}_i \mathbf{P}\|_2}{\widehat{F}}$ so that $\mathbf{Pr}\left[\mathcal{E}\right] = \frac{\epsilon \|\mathbf{A}_i \mathbf{x}\|_2}{\widehat{F}}$. Let $\mathcal{E}_1$ be the event that COUNTSKETCH, AMS, or ESTIMATOR fails so that $\mathbf{Pr}\left[\mathcal{E}_1\right] = \frac{1}{\mathrm{poly}(n)}$ by Lemma A.2, Lemma A.1, and Theorem 2.1. Let $\mathcal{E}_2$ be the event that $\widehat{S} > \widehat{F}\sqrt{\log \frac{1}{\epsilon}}$ so that $\mathbf{Pr}\left[\mathcal{E}_2\right] = \mathcal{O}\left(\epsilon\right)$ by Lemma A.3. Let $\mathcal{E}_3$ be the event that multiple rows $\mathbf{B}_j \mathbf{x}$ exceeding the threshold are observed in the CountSketch data structure and $\mathcal{E}_4$ be the event that $\|\mathbf{B}_i \mathbf{x}\|_2$ exceeds the threshold but is not reported due to noise in the CountSketch data structure. Observe that $\mathcal{E}_3$ and $\mathcal{E}_4$ are essentially two sides of the same coin, where error is incurred due to the inaccuracies of CountSketch.

To analyze $\mathcal{E}_3$, note that row $j \neq i$ can be reported as exceeding the threshold if $\|\mathbf{B}_j \mathbf{x}\|_2 \geq \frac{1}{\epsilon}\widehat{F} - \widehat{F}\sqrt{\log \frac{1}{\epsilon}}$, which occurs with probability at most $\mathcal{O}\left(\frac{\epsilon \|\mathbf{A}_j \mathbf{x}\|_2}{\widehat{F}}\right)$. By a union bound over all rows $j \in [n]$ with $j \neq i$, then $\mathbf{Pr}\left[\mathcal{E}_3\right] = \mathcal{O}\left(\epsilon\right)$.

To analyze $\mathcal{E}_4$, we first condition on $\neg\mathcal{E}_1$ and $\neg\mathcal{E}_2$, so that $\|\mathbf{Bx} - \mathbf{M}\|_2 \leq \widehat{S} \leq \widehat{F}\sqrt{\log \frac{1}{\epsilon}}$. Then by Lemma A.2, the estimate $\widehat{\mathbf{B}_i \mathbf{x}}$ for $\mathbf{B}_i \mathbf{x}$ output by the sampler satisfies

$$\left| \|\mathbf{B}_i \mathbf{x}\|_2 - \left\|\widehat{\mathbf{B}_i \mathbf{x}}\right\|_2 \right| \leq \epsilon \left\|(\mathbf{Bx})_{tail\left(\frac{2}{\epsilon^2}\right)}\right\|_F \leq \epsilon \|\mathbf{Bx} - \mathbf{M}\|_F \leq \epsilon\widehat{S} \leq \epsilon\widehat{F}\sqrt{\log \frac{1}{\epsilon}}.$$

Hence, $\mathcal{E}_4$ can only occur for

$$\frac{1}{\epsilon}\widehat{F} \leq \|\mathbf{B}_i \mathbf{x}\|_2 \leq \frac{1}{\epsilon}\widehat{F} + \epsilon\widehat{F}\sqrt{\log \frac{1}{\epsilon}},$$

which occurs with probability at most $\mathcal{O}\left(\epsilon^2\right)$.

To put things together, $\mathcal{E}$ occurs with probability $\frac{\epsilon \|\mathbf{A}_i \mathbf{x}\|_2}{\widehat{F}}$, in which case the $L_{1,2,d}$ sampler should output $\mathbf{A}_i \mathbf{x}$. However, this may not happen due to any of the events $\mathcal{E}_1$, $\mathcal{E}_2$, $\mathcal{E}_3$, or $\mathcal{E}_4$. Since $\mathbf{Pr}\left[\mathcal{E}_2 \vee \mathcal{E}_3 \mid \mathcal{E}\right] = \mathcal{O}\left(\epsilon\right)$ and $\mathbf{Pr}\left[\mathcal{E}_4\right] = \mathcal{O}\left(\epsilon^2\right)$, then we have $\mathbf{Pr}\left[\mathcal{E}_4 \mid \mathcal{E}\right] = \mathcal{O}\left(\epsilon\right)$. Moreover, $\mathbf{Pr}\left[\mathcal{E}_1\right] = \frac{1}{\mathrm{poly}(n)}$ so that index $i$ is sampled with probability $(1 + \mathcal{O}\left(\epsilon\right))\frac{\|\mathbf{A}_i \mathbf{x}\|_2}{\widehat{F}}$. Finally by Lemma A.2, $\left| \|\mathbf{B}_i \mathbf{x}\|_2 - \left\|\widehat{\mathbf{B}_i \mathbf{x}}\right\|_2 \right| \leq \epsilon\widehat{F}\sqrt{\log \frac{1}{\epsilon}}$ and $\left\|\widehat{\mathbf{B}_i \mathbf{x}}\right\|_2 \geq \frac{1}{\epsilon}\widehat{F}$. Hence $\left\|\widehat{\mathbf{B}_i \mathbf{x}}\right\|_2$ is a $(1 + \epsilon)$ approximation to $\|\mathbf{B}_i \mathbf{x}\|_2$ and therefore, $t_i \left\|\widehat{\mathbf{B}_i \mathbf{x}}\right\|_2$ is a $(1 + \epsilon)$ approximation to $\|\mathbf{A}_i \mathbf{x}\|_2$. □

Thus we have the following full guarantees for our $L_{1,2,d}$ sampler.

**Theorem A.5** *Given $\epsilon > 0$, there exists an algorithm that takes a matrix $\mathbf{A} \in \mathbb{R}^{nd \times d}$, which can be written as $\mathbf{A} = \mathbf{A}_1 \circ \ldots \circ \mathbf{A}_n$, where each $\mathbf{A}_i \in \mathbb{R}^{d \times d}$. After $\mathbf{A}$ is processed, the algorithm is given a query vector $\mathbf{x} \in \mathbb{R}^d$ and outputs a (noisy) vector $\mathbf{A}_i \mathbf{x}$ with probability $(1 \pm \mathcal{O}\left(\epsilon\right))\frac{\|\mathbf{A}_i \mathbf{x}\|_2}{\|\mathbf{Ax}\|_{1,2,d}} + \frac{1}{\mathrm{poly}(n)}$. The algorithm uses $\log \frac{1}{\delta}\cdot\mathrm{nnz}(\mathbf{A}) + \mathrm{poly}\left(d, \frac{1}{\epsilon}, \log n\right)$ time, $\mathcal{O}\left(d\left(\mathrm{poly}\left(\frac{1}{\epsilon}, \log n\right) + \log \frac{1}{\delta}\right)\right)$ bits of space, and succeeds with probability at least $1 - \delta$.*

**Proof :** By Lemma A.4 and Theorem 2.1, then $\|\mathbf{AP}\|_{1,2,d} \leq \widehat{F} \leq 2 \|\mathbf{AP}\|_{1,2,d}$ with high probability and so each vector $\mathbf{A}_i \mathbf{x}$ is sampled with probability $(1 + \epsilon)\frac{\|\mathbf{A}_i \mathbf{x}\|_2}{\|\mathbf{Ax}\|_{1,2,d}} + \frac{1}{\mathrm{poly}(n)}$, conditioned on the sampler succeeding. The probability that the sampler succeeds is $\Theta(\epsilon)$, so the sampler can be repeated $\mathcal{O}\left(\frac{1}{\epsilon}\log n\right)$ times to obtain probability of success at least $1 - \frac{1}{\mathrm{poly}(n)}$. Since each instance of AMS, ESTIMATOR, and COUNTSKETCH use $\log \frac{1}{\delta} \cdot \mathrm{nnz}(\mathbf{A}) + \mathrm{poly}\left(d, \frac{1}{\epsilon}, \log n\right)$ time and $\mathcal{O}\left(d\left(\mathrm{poly}\left(\frac{1}{\epsilon}, \log n\right) + \log \frac{1}{\delta}\right)\right)$ bits of space, then the total time and space complexity follow. □

By adding sketches corresponding to different polynomial degrees, Theorem A.5 implies Theorem 2.2 and Theorem 3.2.

## A.2 LEVERAGE SCORE SAMPLER

Our starting point is the input sparsity time algorithm of (Nelson & Nguyen, 2013) for approximating the leverage scores, which is in turn a modification of (Drineas et al., 2012; Clarkson & Woodruff,

2013) Given an input matrix $\mathbf{A}$, (Nelson & Nguyen, 2013) randomly samples a sparse matrix $\Pi_1$ with $\tilde{\mathcal{O}}\left(\frac{d}{\epsilon^2}\right)$ rows and $\tilde{\mathcal{O}}\left(\frac{1}{\epsilon}\right)$ signs per column, setting the remaining entries to be zero. (Nelson & Nguyen, 2013) maintains $\Pi_1\mathbf{A}$ and post-processing, computes $\mathbf{R}^{-1}$ so that $\Pi_1\mathbf{A}\mathbf{R}^{-1}$ has orthonormal columns. Previous work of (Drineas et al., 2012) had shown that the squared row norms of $\mathbf{A}\mathbf{R}^{-1}$ are $(1+\epsilon)$-approximations to the leverage scores of $\mathbf{A}$. Hence for a JL matrix $\Pi_2$ that gives $(1+\epsilon)$-approximations to the row norms of $\mathbf{A}\mathbf{R}^{-1}$, we can compute $\mathbf{A}(\mathbf{R}^{-1}\Pi_2)$ and output the row norms of $\mathbf{A}\mathbf{R}\Pi_2$ as the approximate leverage scores for each row. Due to the sparsity of $\Pi_1$ and $\Pi_2$, the total runtime is $\tilde{\mathcal{O}}\left(\frac{1}{\epsilon^2}\cdot\text{nnz}(\mathbf{A})\right)$. Computing $\mathbf{R}^{-1}$ takes additional $\tilde{\mathcal{O}}\left(\frac{d^\omega}{\epsilon^2}\right)$ runtime.

Now since the squared row norms of $\mathbf{A}\mathbf{R}^{-1}$ are $(1+\epsilon)$-approximations to the leverage scores of $\mathbf{A}$, it suffices to take the rows of $\mathbf{A}\mathbf{R}^{-1}$ with large squared norms. To that effect, we randomly sample a CountSketch matrix $\mathbf{T}$ and maintain $\mathbf{T}\mathbf{A}$. Once $\mathbf{R}^{-1}$ is computed, we can post-processing right multiply to obtain $\mathbf{T}\mathbf{A}\mathbf{R}^{-1}$, similar to Algorithm 5. It follows that any row of $\mathbf{T}\mathbf{A}\mathbf{R}^{-1}$ that is at least $\frac{1}{200Td}$-heavy (with respect to squared Frobenius norm) has leverage score at least $\frac{1}{100Td}$. Thus we can obtain these rows by querying the CountSketch data structure while using space $\mathcal{O}(Td)$. Due to the sparsity of the CountSketch matrix, the total runtime is $\tilde{\mathcal{O}}\left(\text{nnz}(\mathbf{A})+\frac{d^\omega}{\epsilon^2}\right)$. Finally, (Mahabadi et al., 2020) show that the error guarantee on each reported heavy row required by Theorem 2.3. By reporting the outer products of each of the heavy rows rather than the heavy rows, we obtain Theorem 3.3.

### A.3    APPROXIMATE SGD WITH IMPORTANCE SAMPLING

**Proof of Lemma 2.4:**    For any $t\in[T]$ and $i\in[n]$, $\|\mathbf{A}_i\mathbf{x}_t\|_2^2\geq\frac{1}{100Td}\|\mathbf{A}\mathbf{x}_t\|_F^2$ only if there exists a row in $\mathbf{A}_i$ whose leverage score is at least $\frac{1}{100Td}$, since there are $d$ rows in $\mathbf{A}_i$. Algorithm 2 calculates a 2-approximation to each leverage score and maintains $T$ separate instances of the $L_{1,2,d}$ samplers for any matrix containing a row with approximate leverage score at least $\frac{1}{100Td}$. Thus for these indices $i\in[n]$, we maintain $T$ separate instances of the $L_{1,2,d}$ samplers for $\mathbf{A}_i$ by explicitly maintaining the heavy row.

Otherwise, for all $j\in[\beta]$ so that $h(i)\neq j$ for any index $i\in[n]$ such that $\|\mathbf{A}_i\mathbf{x}_t\|_2^2<\frac{1}{100Td}\|\mathbf{A}\mathbf{x}_t\|_F^2$, we have

$$\sum_{i:h(i)=j}\|\mathbf{A}_i\mathbf{x}_t\|_2^2\leq\frac{1}{100T}\|\mathbf{A}\mathbf{x}_t\|_F^2,$$

with probability at least $\frac{99}{100}$ by Bernstein's inequality and a union bound over $j\in[\beta]$ for $\beta=\Theta(Td)$ with sufficiently high constant. Intuitively, by excluding the hash indices containing "heavy" matrices, the remaining hash indices contain only a small fraction of the mass with high probability. Then the probability that any $j\in[\beta]$ with $\sum_{i:h(i)=j}\|\mathbf{A}_i\mathbf{x}_t\|_2\leq\frac{1}{10T}\|\mathbf{A}\mathbf{x}_t\|_{1,2,d}$ is sampled more than once is at most $\frac{1}{100T}$ for any $t\in[T]$ provided there is no row in any $\mathbf{A}_i$ with $h(i)=j$ whose $\ell_2$ leverage score is at least $\frac{1}{100Td}$. Thus, the probability that some bucket $j\in[\beta]$ is sampled twice across $T$ steps is at most $\frac{\beta}{(100T)^2}\leq\frac{1}{100}$.

In summary, we maintain $T$ separate instances of $L_{1,2,d}$ samplers for the heavy matrices and one $L_{1,2,d}$ sampler for each hash index that does not contain a heavy matrix. With probability at least $\frac{98}{100}$, any hash index not containing a heavy matrix is sampled only once, so each time $t\in[T]$ has access to a fresh $L_{1,2,d}$ sampler.    □

## B    EMPIRICAL EVALUATIONS

We again emphasize that our primary contribution is the theoretical design of a nearly input sparsity time streaming algorithm that simulates the optimal importance sampling distribution for variance reduction in stochastic gradient descent without computing the full gradient. Thus our theory is optimized to minimize the number of SGD iterations without asymptotic wall-clock time penalties; we do not attempt to further optimize wall-clock runtimes. Nevertheless, in this section we implement a scaled-down version of our algorithm and compare its performance across multiple iterations on large-scale real world data sets to SGD with uniform sampling on both linear regression and

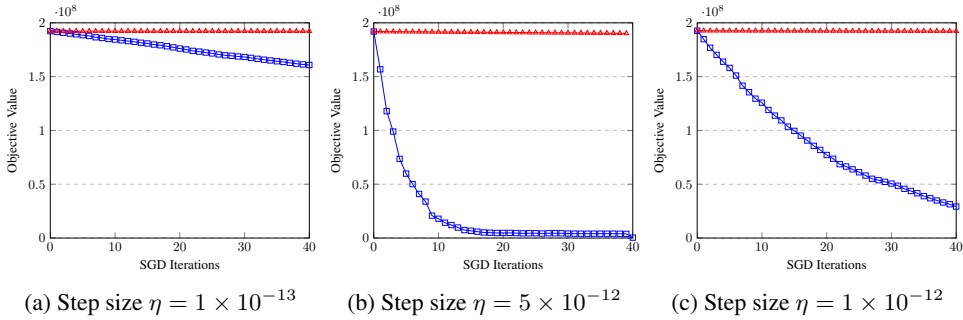

Fig. 3: Comparison of importance sampling (in blue squares) and uniform sampling (in red triangles) over various step-sizes for linear regression on CIFAR-10.

support-vector machines (SVMs). Because most rows have roughly uniformly small leverage scores in real-world data, we assume that no bucket contains a row with a significantly large leverage score and thus the implementation of our importance sampling algorithm does not create multiple samplers for any buckets. By similar reasoning, our implementation uniformly samples a number of indices $i$ and estimates $\|\mathbf{A}\mathbf{x}\|_{1,2,d} = \sum_j \|\mathbf{A}_j\mathbf{x}\|_{1,2,d}$ by scaling up $\|\mathbf{A}_i\mathbf{x}\|_{1,2,d}$. Observe that although these simplifications to our algorithm decreases the wall-clock running time and the total space used by our algorithm, they only decrease the quality of our solution for each SGD iteration. Nevertheless, our implementations significantly improve upon SGD with uniform sampling. The experiments in this section were performed on a Dell Inspiron 15-7579 device with an Intel Core i7-7500U dual core processor, clocked at 2.70 GHz and 2.90 GHz, in contrast to the logistic regression experiments that were performed on a GPU.

**Linear Regression.** We performed linear regression on the CIFAR-10 dataset to compare the performance of our importance sampling algorithm to the uniform sampling SGD algorithm. We trained using a data batch of 100000 points and 3072 features and tested the performance on a separate batch of data points. We aggregated the objective values across 10 separate instances. Each instance generated a random starting location as an initial position for both importance sampling and uniform sampling. We then ran 40 iterations of SGD for each algorithm and observed the objective value on the test data for each of these iterations. Finally, we computed the average performance on each iteration across these 10 separate instances. As we ran our algorithm for 40 iterations, we created 1600 buckets that partitioned the data values for the importance sampling algorithm.

The sampled gradients were generally large in magnitude for both importance sampling and uniform sampling and thus we required small step-size. For step-sizes $\eta = 1 \times 10^{-13}$, $\eta = 5 \times 10^{-12}$, and $\eta = 1 \times 10^{-12}$, the objective value of the solution output by our importance sampling algorithm quickly and significantly improved over the objective value of the solution output by uniform sampling. Our algorithm performance is much more sensitive to the choice of larger step-sizes, as choices of step-sizes larger than $5 \times 10^{-11}$ generally caused the importance sampling algorithm to diverge, while the uniform sampling algorithm still slowly converged. We give our results in Figure 3.

**Support-Vector Machines.** We also compared the performance of our importance sampling algorithm to the uniform sampling SGD algorithm using support-vector machines (SVM) on the a9a Adult data set collected by UCI and retrieved from LibSVM (Chang & Lin, 2011). The features correspond to responses from the 1994 Census database and the prediction task is to determine whether a person makes over 50K USD a year. We trained using a data batch of 32581 points and 123 features and tested the performance on a separate batch of 16281 data points. We assume the data is not linearly separable and thus use the hinge loss function so that we aim to minimize $\frac{1}{n}\sum_{i=1}^{n} \max(0, 1 - y_i(\mathbf{w} \cdot \mathbf{X}_i - b)) + \lambda \|\mathbf{w}\|_2^2$, where $\mathbf{X}$ is the data matrix, $y_i$ is the corresponding label, and $\mathbf{w}$ is the desired maximum-margin hyperplane. For each evaluation, we generated 10 random initial positions shared for both importance sampling and uniform sampling. We then ran 75 iterations of SGD for each algorithm, creating 1125 buckets for the importance sampling algorithm and computed the average performance on each iteration across these 5 separate instances.

The sampled gradients were generally smaller than those from linear regression on CIFAR-10 and thus we were able to choose significantly larger step-sizes. Nevertheless, our algorithm performance was sensitive to both the step-size and the regularization parameter. For step-sizes $\eta = 0.25$, $\eta = 0.5$ and regularization parameters $\lambda = 0$, $\lambda = 0.001$ and $\lambda = 0.0001$, the objective value of the solution output by our importance sampling algorithm quickly and significantly improved over the objective value of the solution output by uniform sampling. We give our results in Figure 5. Our algorithm performance degraded with larger values of $\lambda$, as well step-sizes larger than $\eta = 1$.

We also compared step-size $\eta = 1$ and regularization parameters $\lambda = 0$, $\lambda = 0.001$ and $\lambda = 0.0001$ with a hybrid sampling scheme that selects the better gradient between importance sampling and uniform sampling at each step, as well as a hybrid sampling scheme that uses a few steps of importance sampling, followed by uniform sampling in the remaining steps. Our experiments show that the hybrid sampling algorithms perform better at the beginning and thus our importance sampling algorithm may be used *in conjunction* with existing techniques in offline settings to accelerate SGD. Surprisingly, the hybrid sampling algorithms do not necessarily remain better than our importance sampling algorithm thus indicating that even if uniform sampling were run for a significantly larger number of iterations, its performance may not exceed our importance sampling algorithm. We give our results in Figure 6.

Finally, we compare wall-clock times of each of the aforementioned sampling schemes with step-size $\eta = 1$ and regularization 0 across 100 iterations. Our results in Figure 4 show that as expected, uniform sampling has the fastest running time. However, each iteration of importance sampling takes about 15 iterations of uniform sampling, which empirically shows that even using wall-clock times for comparison, rather than total number of SGD iterations, the performance of our importance sampling algorithm *still* surpasses that of uniform sampling. Moreover, the runtime experiments reveal the main bottleneck of our experiments: each of the 100 iterations took approximately 70 seconds on average after including the evaluation of the objective on each gradient step.

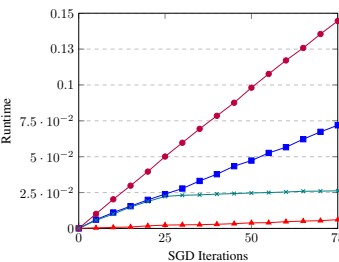

Fig. 4: Runtimes comparison for SVM on a9a Adult dataset from LibSVM/UCI with step size $1.0$ and regularization 0, averaged over 100 iterations: importance sampling (in blue squares), uniform sampling (in red triangles), hybrid sampling that chooses the better gradient at each step (in purple circles), and hybrid sampling that performs 25 steps of importance sampling followed by uniform sampling (in teal X's). By comparison, the average *total* runtime over 100 iterations was 72.5504 seconds, including the computation of the scores.

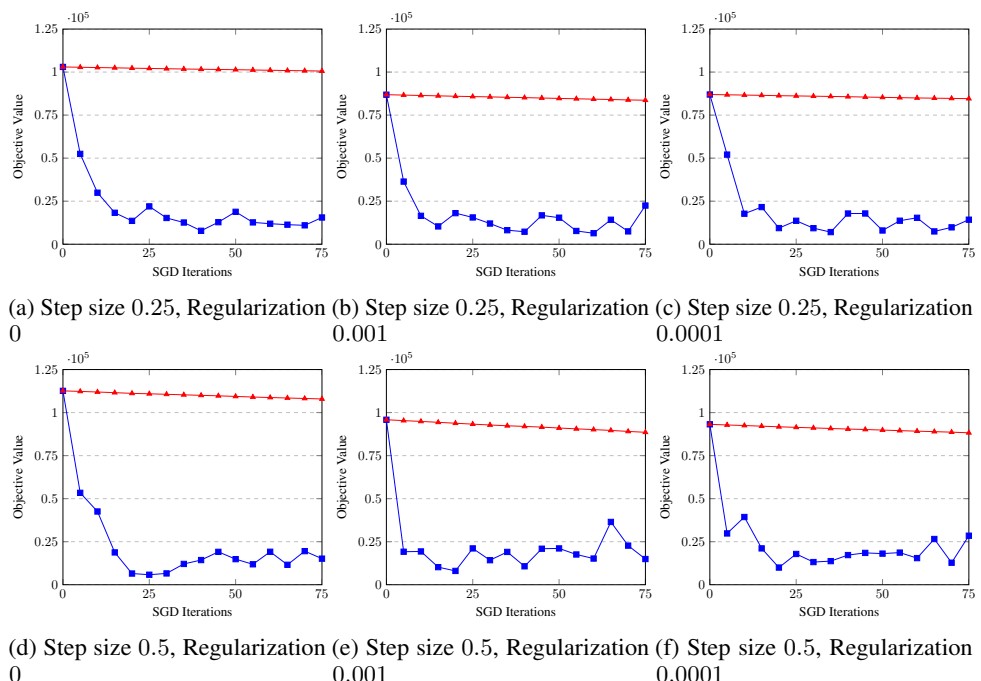

(a) Step size 0.25, Regularization 0

(b) Step size 0.25, Regularization 0.001

(c) Step size 0.25, Regularization 0.0001

(d) Step size 0.5, Regularization 0

(e) Step size 0.5, Regularization 0.001

(f) Step size 0.5, Regularization 0.0001

Fig. 5: Comparison of importance sampling (in blue squares) and uniform sampling (in red triangles) over various step-sizes for SVM on a9a Adult dataset from LibSVM/UCI, averaging across ten iterations, averaged over 10 iterations.

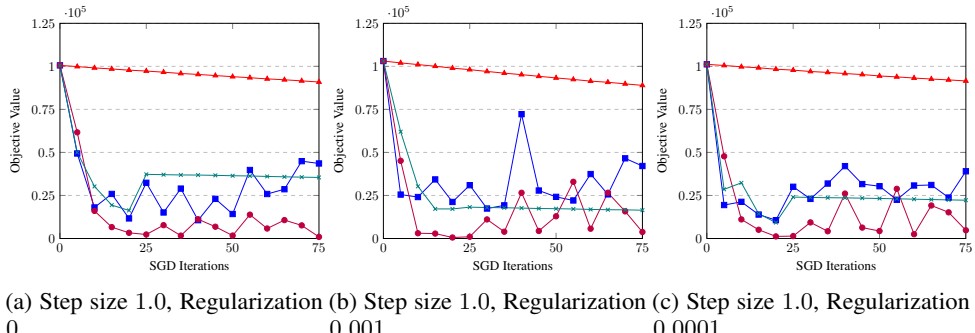

(a) Step size 1.0, Regularization 0

(b) Step size 1.0, Regularization 0.001

(c) Step size 1.0, Regularization 0.0001

Fig. 6: Comparison of importance sampling (in blue squares), uniform sampling (in red triangles), hybrid sampling that chooses the better gradient at each step (in purple circles), and hybrid sampling that performs 25 steps of importance sampling followed by uniform sampling (in teal X's) over various step-sizes for SVM on a9a Adult dataset from LibSVM/UCI, averaged over 10 iterations

