# OpenReview forum: "Adaptive Single-Pass Stochastic Gradient Descent in Input Sparsity Time"
_ICLR.cc/2021/Conference — Reject_

### Official Review · AnonReviewer1 · 2020-10-29
**Interesting theoretical application of sketching; clarity issues**

**Rating:** 6
**Confidence:** 3

**Review:**

Summary:
This paper develops an efficient streaming algorithm to approximate the optimal importance sampling weights for variance reduction in finite-sum SGD. The optimal weights are proportional to each sample's gradient norm; this work uses AMS-like moment estimation to sketch gradient norms which take the form of a bounded-degree polynomial, in time linear in the input sparsity and polynomial in the dimension d, iteration count T, and the log of the number of the samples n. A second-order analogue is derived for approximating optimal importance weights for sampling the Hessian. Some experiments are shown with more simplistic importance weight estimators (not the proposed algorithm), to demonstrate the advantage over uniform sampling.

Pros:
- To my knowledge, this is a novel way to bridge the tools of moment sketching/leverage score sampling with the problems and subproblems in large-scale stochastic optimization.
- The technical contribution is overall solid and interesting, despite some caveats pointed out below. The decoupling of T from nnz(A) is striking.

Cons:
- This algorithm is somewhat far from being useful (as acknowledged in the first paragraph of Appendix B), despite the last sentence in the abstract. The unconventional regime of d << T << n seems necessary to experience speedups; d-by-d matrix inversions are needed the compute the leverage scores. The "Estimator" routine has to maintain a d^(deg f)-sized tensor. This doesn't count against the paper, but it should thus be evaluated as a theoretical work.
- My primary concern, and the reason I believe this work could use a round of significant revisions before publication, is with clarity. The relationship with prior work could stand to be clearer; the results are presented in a rather unconventional way (see comments below).

Detailed comments:
- A source of confusion: at the beginning of Section 2, under "L_2 polynomial inner product sketch", it's claimed that we require a constant-factor approximation to the norm of the population gradient (sum inside the norm). Then, it seems that all theorems in the main paper are concerned with a sum on the outside, of the norms squared. Then, in the appendix, where the claims are proven, this issue goes away, as the _{1,2,d} norm is a summation of the non-squared norms. A clarification would be appreciated, if there was some reason to switch between norms and squared norms in the main paper, or if some subset of these are typos. Given the above, the proof of Theorem 1.1 in the main paper is hard to follow, in how it uses Theorem 2.2 (top of page 6).
- "The generalization to a L_2 polynomial inner product sampler follows immediately." Though it might be straightforward, it's probably best to include the full analysis in the appendix, and show the dependences on the degree of the polynomial. (If I understand correctly, these are straightforward analogues in the same way that the tensor procedure in Alg. 1 is a straightforward analogue of scalar moment sketching.)
- There is no explicit convergence theorem accompanying Algorithm 2; only that it performs T steps of gradient descent. However, this is a minor point, as it should suffice to push the variance bound through the analysis found in the cited work [Zhao & Zhang '15]. Including an explicit convergence rate would be helpful, since the paper culminates in an optimization algorithm.
- Specific discussions on the cited works on variance reduction, with comparisons of assumptions and convergence rates, would be helpful for placing this work in the context of those results.
- The paper derives much of its technical content from [Mahabadi et al. '20]; the exposition would benefit from more specific pointers to the theorems and lemmas from that work, as well as where this paper goes beyond those results, as opposed to direct applications.
- There is also no explicit theorem accompanying the second-order algorithm. The black-box oracle model with Hessian estimators is problematic, since their inverses may not be usable, so approximation guarantees don't give convergence rates for free. See [1,2] for some work on getting stochastic second-order optimization methods to converge.

[1] Erdogdu & Montanari, "Convergence Rates of Sub-Sampled Newton Methods".
[2] Agarwal et al., "Second-Order Stochastic Optimization for Machine Learning in Linear Time".


*** post-response ***

Thanks for the response and extensive modifications. I have increased my score to "weak accept" to reflect the improved clarity. I think the paper remains on the borderline, for the following reasons:

- I still doubt the practicality of this algorithm in any realistic settings. The experiments demonstrate that one can gain on the variance term from importance sampling in SGD, but not the practicality of the moment-sketching one-pass estimator of multiple gradient queries on practical problem sizes. Thus I disagree with the "vastly superior" characterization; it is well-known that the gradient estimator this paper seeks to efficiently approximate is a good choice; a convincing proof-of-concept is that it's useful to use sketching to efficiently approximate it (whether it's worth the computational overhead + approximation error). The wall-clock time experiment is independently interesting, perhaps showing that you don't even need the sketching ideas from this paper to benefit from importance-sampled stochastic gradients. That said, my evaluation discounts the experimental part and evaluates this as a purely theoretical work.
- The theoretical work is an application of [Mahabadi et al. '20], without discernible major theoretical innovations. Though, in my opinion, the application to SGD makes it creative and relevant enough for publication, despite practicality concerns.
- Some clarity issues remain, rendering the paper hard to digest: squared-norms in main paper vs. norms in appendix, hidden dependences on p, "follows immediately". After thinking about it, the "implicitly computed" tensor contractions in the comment to R4 makes sense, but this should probably be pointed out clearly in the paper, if I correctly understand that it removes all d^p factors from the analysis.

---

> ### Author Response · Authors · 2020-11-25
> **Response to Reviewer 1**
>
> We thank the reviewer for their detailed comments. We have reworked our manuscript to emphasis clarity. We agree that a convergence guarantee should be included, and we have incorporated such a statement into our revised version. Moreover, we have included statements on the convergence guarantee and compared the variance of importance sampling versus uniform sampling in SGD. We have also added discussion in the beginning of the supplementary material to explain the context switch from the $L_2$ norm sampling to the $L_{1,2,d}$ norm sampling.
>
> Finally, we have expanded our empirical evaluations to include experiments comparing our algorithm to standard SGD in wall-clock time. Since our results in the revision version demonstrate vastly superior performance of our algorithm even in wall-clock time, even without attempts to optimize our running time, we hope our additional results serve as compelling evidence that our initial proof-of-concept is an important step toward practical application.
>
> We agree with the reviewer that we do not study specific Hessian based methods. We believe that rather than give specific convergence guarantees, the central message of our paper should be a one-pass algorithm that allows for multiple rounds of optimal importance sampling through variance reduction, for both first-order and second-order optimization.

---

### Official Review · AnonReviewer4 · 2020-10-29
**Interesting theory algorithm, but questions about practical implications**

**Rating:** 6
**Confidence:** 3

**Review:**

The present paper shows how to perform variance reduction for SGD (as well as stochastic second-order methods) in a streaming setting. Their algorithms generally run in roughly O(nnz + poly(d)) time, where nnz is the number of non-zero entries in the data and poly(d) is an unspecified polynomial in the input dimension (I believe it is often d^2, but I hope the authors can clarify; see questions below).

Streaming variance reduction is an important practical problem that could potentially speed up model training substantially. As far as I can tell, the theoretical algorithm is correct and the analysis, while a bit terse, is overall well-presented (especially given space constraints). My main concerns are around whether the algorithm really delivers in practice.

I have two concerns along this line. The first is that the theory requires the gradient of the loss to be of the form f(<a,x>) * a, where f is a polynomial. I have trouble thinking of cases where this is the case (even in logistic regression and SVMs, f is not a polynomial; and for neural nets the loss doesn't even have this form). It would help if the authors could clarify this point, since I didn't see it explained in the experiments on logistic regression.

My second concern is over whether the algorithm is really "nearly linear-time" in a practically meaningful sense, given that there is a poly(d) component to the running time, and the exponent in d is not specified. Looking through the actual algorithm and analysis, I believe it is O(d^p) where p is the degree of f, which if we take a second-order approximation to f means at least O(d^2). In many applications this would be infeasible (many models have d > n, so d^2 is worse than n*d). Figure 1 suggests the running time is indeed much slower compared to vanilla SGD. I'd like it if the authors could clarify the actual exponent in the running time, and also present a version of Figure 1 plotting accuracy against wall clock time, which seems like the more relevant comparison between the two algorithms.

I currently give the paper a weak reject, both because I am not currently convinced of the practical relevance of the algorithm, and because I feel the paper somewhat brushes these practical issues under the rug, which could be confusing to readers. However, I would increase my score if the author response to my questions is able to sway me on the practical import question.

---

> ### Author Response · Authors · 2020-11-25
> **Response to Reviewer 4**
>
> We thank the reviewer for their detailed feedback. We remark that the runtime of our algorithm does indeed include a dependency on $O(d^2)$, due to matrix multiplication. Although we also require a $p$-wise tensor, these terms can be implicitly computed when necessary, thus reducing to the input sparsity runtime.
>
> We agree with the reviewer that theory requires the gradient of the loss to be of the form $f(\langle a,x\rangle)\cdot a$. Thus our results achieve optimization for the class of polynomials. Moreover, we achieve results for the class of functions whose gradients are well-approximated by polynomials. For loss functions such as logistic regression, we can expand the Taylor series and compute their gradients. Although it is far from clear that such an approximation would provide accurate guarantees, our initial experiments indicate that our algorithm performs well in practice.
>
> We also agree that although our analysis provably guarantees better performance than standard SGD per iteration, the more interesting comparison is over wall-clock time. Along the same veins as our response to Reviewer 2, we have also expanded our empirical evaluations to include experiments comparing our algorithm to standard SGD in wall-clock time. Our results in the revision version demonstrate vastly superior performance of our algorithm even in wall-clock time, even without attempts to optimize our running time.  We therefore hope that our results will be the first step toward broad practical application.

---

### Official Review · AnonReviewer3 · 2020-10-30
**Weak reject recommendation**

**Rating:** 5
**Confidence:** 3

**Review:**

#### Problem statement

Consider the scenario where we wish to compute the gradient of
$$ F(x) = 1/n\sum_i f_i(x)$$
which is given by
$$ \nabla F(x) = 1/n\sum_i \nabla f_i(x). $$

Stochastic gradient descent (SGD) is a computationally efficient alternative to full -blown gradient descent (GD), which gives an unbiased estimator of the gradient, we pick $i$ uniformly at random and compute $\nabla f_i$. The estimator might have huge variance, hence variance reduction approaches are studied, which sample each $f_i$ with importance weights probability proportional to the (normalized) squared norm of its gradient. But  computing this probability naively is as costly as computing $\nabla F$.

This paper proposes a more sophisticated approach to computing importance weights, in the setting where the functions $f_i$ have a special form (essentially polynomials in the inner product with a fixed vector). There is a matrix $A$ where each row $a_i$ defines a function $f_i(x) = P(<a_i,x> )$ for some univariate polynomial $P(t)$, so that its gradient is of the form $\nabla f_i = P'(<a_i,x>)a_i$. The punchline of this paper is that it can compute a reasonable approximation to the true importance weights in time linear in $O(nnz(A))$ where $nnz(A)$ is the number of non-zero entries in $A$.

#### Pros

* The general problem is clearly important and interesting. Conditioned on the class of functions  (more about that later), the result they get is strong.

* The individual technical ingredients such as the  polynomial inner product sampler and the leverage score sampler are interesting in their own right.

* The paper seems mathematically sound and comfortably above bar at a technical level.

#### Cons

My main concerns are with the presentation.

* Let us start with the class of functions that can be handled. The paper defines this class as functions such that
$$\nabla f_i(x) = p(<a_i, x>)a_i$$
To my knowledge, this is just polynomails in the inner product:
$$f_i(x) = P(<a_i, x>) \ \text{where} \ P'(t) = p(t).$$
If so, stating it in the form of a gradient equation is a little strange.

* There needs to be some discussion of why this is an interesting class of functions in this setting, what it covers, and what it doesn't. . The authors mention applications to neural nets at some point in the Introduction. It is unclear how their results apply to neural nets (or even to logistic regression, on which they report experiments), given $P$ is polynomial in the inner product.

* In general, the paper is very dense. I was unable to get a sense of what the technical novelty of the paper is, and where the authors  are putting together known ideas from previous work.  (See suggestions below).

#### Suggestions

* It might be worth pointing out that in the simplest case when $f_i = < a_i, x>$ is trivial since you only need the sum of the rows. For the case $f_i = <a_i, x>^2$, it seems related to SVD decomposition. So is it the degree $3$ case the simplest case you where you need to do something new? Is there a simple explanation for how your algorithm works in this case? Is there a connection to the $L_p$ sampling problem?

#### Summary of Recommendation

At this point, my concerns are mainly with the presentation of the paper, not its content. Whereas the problem it tackles is of broad interest, in its current form, the paper will not appeal to a broad ML audience,

---

> ### Author Response · Authors · 2020-11-25
> **Response to Reviewer 3**
>
> We thank the reviewer for their detailed comments. We emphasize that our results are stated in terms of the gradient of $f$, so that any function whose gradient is well-approximated by a polynomial can be optimized under our setting. By contrast, note that a function that is well-approximated by a polynomial may not have a gradient that is well-approximated by a polynomial. We thus obtain results for neural nets whose loss functions have gradients that can be well-approximated by a polynomial.
>
> Per the suggestion of the reviewer, we have also incorporated additional intuition for our algorithm. The key novelty we use is to generalize the relatively new adaptive sampling data structure given by [Mahabadi, et. al. 2020], which is indeed based on $L_p$ sampling, but also allows for adaptive queries while only requiring a single pass over the data. The main point is that the adaptive sampling data structure can read the vectors in a single pass, but then simulate importance sampling after forcing all the vectors to rotate post-processing. For our purposes, the post-processing corresponds to each new location in the SGD iterate.
>
> In the revised version, we have also expanded the introduction to include a discussion on the convergence rate of our algorithm, as well as included specific examples when the variance reduction of importance sampling over the uniform sampling of SGD is significant. We hope the reworked presentation will make the paper more accessible to a general ML audience.

---

### Official Review · AnonReviewer2 · 2020-11-03
**Optimal importance sampling technique for finite Sum-SGD**

**Rating:** 6
**Confidence:** 3

**Review:**

Here is the review of the article: ``Adaptive Single-Pass Stochastic Gradient Descent in Input Sparsity Time''.

**Summary**

The authors proposed a method consisting in proposing the best-sampling schemes to reduce the variance of the Stochastic Gradient Descent algorithm in the finite sum setting.

More precisely, they resort to importance sampling techniques to lower (significantly) the variance of the SGD procedure. However, as a perfectly tuned variance reduced scheme would lead to the same complexity as Gradient descent, they try to estimate for each sample the importance of each gradient.

The main results consists in giving a proof of how works this sampling techniques and why it avoids a $T \cdot nnz(A)$ running time (the one of full GD). Roughly speaking, the algorithm rests on the following building blocks:

(i) The ability to estimate $\sum{\|f(\langle a_i,x \rangle)\cdot a_i\|_2^2}$ efficiently;

(ii) The ability to solve the problem at each step efficiently in $nnz(A)$;

(iii) The ability to circumvent the use of the (ii) $T$-times by separating the data set into $T\cdot d$ buckets and apply a coarse-grained version of (ii).



**Clarity**

Despite the relative complexity of the procedure the paper is relatively comprehensive. The goals are clearly stated and the paper reads rather well when we have understood the purpose of each block. However, it will be very useful for the reader to have an overview understanding of how the algorithm works at the end of the introduction. I cannot yet imagine how the authors can modified the current version to make it more comprehensive, but it is sure that the complexity of the procedure is a real barrier for the reader.

**Quality**

All the results are clearly stated, and I have nothing to say about the veracity of the theorems that seem to be properly referenced and proved (yet I did not have a precise look at the proofs of the article).

**Originality**

I am not a specialist of this importance sampling literature. However, this approach of trying to estimate the best sampling procedure seems quite new. I will come back to possible and proper comparison with the literature in the {\bfseries Comments} section.

**Comments**

*General Comments.* Below the general comments and ways to improve the quality of the article.



**Improve the clarity.** First, some comments regarding the clarity and the paper have been given earlier in the properly named section. As a rather technical and complex article, it is very important to tell to the reader a clear story of how the different blocks build the algorithm. Moreover, I really wonder if the target of such a article is really the ICLR conference. 8 pages do not seem enough to explain the rich ideas of your paper and detail its technical ideas.

**Proper comparison and randomized Kaczmarz algorithm.** Second, and once again, I am not a specialist of the literature, but the referencing on importance sampling SGD does not seem very complete. I would have very liked the article to be compared to what I know from the field, and in my mind: A randomized Kaczmarz algorithm with exponential convergence (Thomas Strohmer, Roman Vershynin) and Stochastic Gradient Descent, Weighted Sampling, and the Randomized Kaczmarz algorithm (Deanna Needell, Nathan Srebro and Rachel Ward) are interesting comparisons to have in mind when it comes to importance sampling for SGD. How does you algorithm compare to them? Is this a computational explanation of how to perform SGD properly without estimating the magnitude of the ligns of the matrix to invert ?

**More experiments.** I know that it is difficult to give a new algorithm, prove that it converges to the right distribution and extensively test it. However, I really think that I will be convinced only if the experiments on real or synthetic examples show a nice (regarding time!) behavior of your algorithm. Indeed, as shown in your example there is a trade-off between the fact that you need more time than SGD on the one hand but lower the variance on the other hand. How can I see this in practice is very important and for now, I am not totally convinced by your simulations. Comparing this to invert largely overparametrized linear system could be a very good benchmark !



*Minor Comments*


The title in itself does not seem to reflect the content on the paper. Indeed, the word "adaptive" is quite confusing (even if I see that it is adaptive to the magnitude of the gradient). Moreover, I would really stress on the fact that it is an importance sampling technique for finite sum SGD.

I really would like that the authors give an intuition for the theorem 2.1 which is the principal building block of the paper. Is this new ? How is it constructed ?

It seems that at the end of page 1 and beginning of page 2 there is a mistake in the variance. I would expect a $n^2$ in front of $\|\nabla F(x_t)\|^2$ in the $\sigma_t^2$ expressions.

Typo: there are several confusion with $p_{i,t}$ and $p_{i_{t}}$ and $p_{i_{t}, t}$

---

> ### Author Response · Authors · 2020-11-25
> **Response to Reviewer 2**
>
> We thank the reviewer for the positive assessment of the paper. The advantages of our algorithm over existing approaches such as [Strohmer and Vershynin, 2007] and [Needell, et. al., 2016] is that our sampling approach only requires a single pass over the data. Subsequent SGD iterations still obtain the optimal variance without requiring additional passes over the data. The key technique we use to achieve this property is based on the relatively new adaptive sampling data structure given by [Mahabadi, et. al. 2020] that also allows adaptive importance sampling while only requiring a single pass over the data, albeit in a very different context. We thus retained the "adaptive sampling" terminology in our title as to reflect the importance of the paradigm of this data structure to our work (though we would not be averse to any other title suggestions).
>
> We agree that although our analysis provably guarantees better performance than standard SGD per iteration, the more interesting comparison is over wall-clock time. We have therefore expanded our empirical evaluations to include experiments comparing our algorithm to standard SGD in wall-clock time. Our results in the revision version demonstrate vastly superior performance of our algorithm even in wall-clock time, even without attempts to optimize our running time.
>
> We thank the reviewer for noticing the typo for the variance expressions. We also thank the reviewer for raising concern about the notation $p_{i,t}$ compared to $p_{i_t,t}$. Specifically, $p_{i,t}$ is the probability of sampling row $i$ at time $t$, while $i_t$ is the row that is sampled at time $t$ and $p_{i_t,t}$ of sampling this row. We have explicitly clarified these notations.

---

### Decision · Program_Chairs · 2021-01-07
**Final Decision**

**Decision:**

Reject

**Comment:**

While this paper was received pretty well, especially after the revision, reviewers still find it borderline and request further revisions which we cannot check in this short review cycle. Therefore, we encourage the authors to improve the paper and resubmit to a future venue. In particular, please take into account the reviewers' comment to improve the clarity of the paper. Particularly it is critical to clarify the function class you are working with (essentially polynomials) more clearly than what you currently do (i.e., your current gradient definition). It would be helpful for future work to clearly state that this function class is a shortcoming of your work, and that an interesting direction is to extend this to natural function classes in ML (e.g., logistic loss).

---

> ### Author Response · Authors · 2021-02-23
> **Incorporation of Reviewer Feedback**
>
> We thank the ICLR 2021 Conference Program Chairs for a detailed summary of the reviewers' remarks. For completeness, we offer the following response summarizing our incorporation of the reviewer feedback to improve future versions of our manuscript.
>
> We have significantly reworked our manuscript to handle sparse inputs to simultaneously address various regimes of $n$ and $d$. Namely, for $n\ll d$, each row often has a small number of non-zero entries so that the true input size is significantly smaller than $nd$. Similarly for $d\ll n$, a large number of rows often will not be selected by importance sampling so that we should not repeatedly examine these rows. We updated the analysis of our algorithm to capture these observations and thus give a runtime in terms of row-sparsity that improves upon our previous poly(d) factors.
>
> We have also significantly improved our manuscript to emphasize clarity, particularly in regard to the class of functions that are handled by our approach (polynomials with non-negative coefficients).  We have incorporated such a statement into our revised version, comparing the variance of our importance sampling approach versus uniform sampling in SGD. We have also improved notational clarity.
>
> In addition, we have expanded our empirical evaluations to include experiments comparing our algorithm to standard SGD in wall-clock time. Since our results in the revision version demonstrate vastly superior performance of our algorithm even in wall-clock time, even without attempts to optimize our running time, we hope our additional results serve as compelling evidence that our initial proof-of-concept is an important step toward practical application.